

**Hydrological control of large hurricane-induced lahars: evidences from rainfall,**
**seismic and video monitoring**
Lucia Capra[1], Velio Coviello[1], Lorenzo Borselli[2], Víctor-Hugo Márquez-Ramírez[1], Raul
Arámbula-Mendoza[3]
*[1] Centro de Geociencias, Universidad Nacional Autónoma de México (UNAM), Campus*
*Juriquilla, Queretaro, México*
*[2] Instituto de Geología, Universidad Autónoma de San Luis Potosí, San Luis Potosí,*
*México*
*[3] Centro Universitario de Estudios e Investigaciones en Vulcanología (CUEIV),*
*Universidad de Colima, Colima, México.*
**Abstract**
The Volcán de Colima, one of the most active volcanoes in Mexico, is commonly affected
by tropical rains related to hurricanes that form over the Pacific Ocean. In 2001, 2013 and
2016 hurricanes Jova, Manuel and Patricia, respectively, promoted tropical storms that
accumulated up to 400 mm of rain in 36 hrs, with maximum intensities of 50 mm/hrs.
Effects were devastating, with the formation of multiple lahars along La Lumbre and
Montegrande ravines, which are the most active channels in sediment delivery on the S-SW
flank of the volcano. Deep erosion along the river channels and several landslides at their
side were observed, and damages to bridges and paved roads for the arrival of block-rich
fronts resulted in the distal reach of the ravines. Based on data from real-time monitoring
(including images, seismic records and rainfall data), the temporal sequence of these events
is reconstructed and analyzed with respect to the rainfall characteristics and the
hydrological response of the watersheds based on rainfall/infiltration numerical simulation.





For the studied events, lahars occurred after 5-6 hours since rainfall started, lasted several
hours and were characterized by several pulses with block-rich fronts and a maximum flow
discharge of 900 m$^3$/s. Rainfall/infiltration simulations were performer with the Flo-2D
code using the SCS-Curve number infiltration model. Results show different behaviors for
the arrival times of the first lahar pulses that correlate with the catchment's peak discharge
for La Lumbre ravine and with the peaks in rainfall intensity for Montegrande ravine. This
different behavior is strictly related to the area and shape of these two watersheds.
Nerveless, for all the analyzed cases, the largest lahar pulse always corresponds with the
last one and correlates with the maximum peak discharge of these catchments. Data here
presented show that main pulses within a lahar are not randomly distributed in time, and
they can be correlated with rainfall peak intensity and/or watershed discharge, depending
on the watershed area and shape. This outcome has important implications for hazard
assessment during extreme hydro-meteorological events since it could help in real-time
alert. A stormwater was here designed based on the rainfall time distribution of hurricanes
Manuel and Patricia and, in case on available weather forecasts, it can be used to run
simulations prior to the event, and have an estimation of the time arrivals of main pulses,
usually characterized by block-rich fronts that are responsible of damage to infrastructures
and loss of goods and lives.

***Keywords**: lahar, hurricane, rainfall/infiltration simulation, Volcán de Colima, Mexico.*

**1. Introduction**



In past recent years hurricanes have had catastrophic effects on volcanoes of the
world triggering lahars (sediment-water gravity-driven flows on volcanoes). One of the
most recent episode is represented by the 2009 Hurricane Ida in El Salvador that caused
several landslides and debris flows from the Chichontepec volcano, killing 124 people, or
by the 1998 Hurricane Mitch that triggered the collapse of a small portion of the inactive
Casita volcano, originating a landslide that suddenly transformed into a lahar that
devastated several towns and killed 2000 people (Van Wyk Vries et al., 2000; Scott et al.,
2005). A similar event was observed in 2005 when tropical storm Stan triggered landslides
and debris flows from the Toliman Volcano (Guatemala), causing more than 400 fatalities
at Panabaj community (Sheridan et al., 2007). Other examples can be found at the
volcanoes Pinatubo (Philippines), Merapi and Semeru (Indonesia), Soufriére (Montserrat),
Mt. Ruapehu (New Zealand), where tropical storms and heavy rainfall seasons have
triggered high-frequency lahar events (Umbal and Rodolfo, 1996; Cronin et al., 1997;
Lavigne et al., 2000; Lavigne and Thouret, 2002; Barclay et al., 2007; Dumaisnil et al.,
2010; Doyle et al., 2010, de Bélizal et al., 2013).
Volcán de Colima (19°31'N, 103°37' W, 3860 m a.s.l., Fig. 1), one of the most
active volcanoes in Mexico, is periodically exposed to intense seasonal rainfalls that are
responsible for the occurrence of lahars from June to late October (Davila et al., 2007;
Capra et al., 2010). Rain-triggered lahars represent a very common process during the rainy
season (June-October) at Volcán de Colima (Davila et al., 2007; Capra et al., 2010;
Vazquez et al., 2016a). They usually affect areas as much as 15 km from the summit of the
volcano, with resulting damage to bridges and electric power towers (Capra et al., 2010),
and are more frequent just after eruptive episodes such as dome collapse emplacing block-

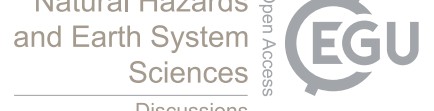



and-ash flow deposits (Davila et al., 2007; Vázquez et al., 2016b). Several hurricanes
commonly hit the Pacific Coast each year and proceed inland as tropical rainstorm reaching
the Volcán de Colima area. In particular, on 2011, 2013 and 2015 Jova, Manuel and
Patricia hurricane respectively triggered long-lasting lahars along main ravines, causing
several damages on roads and bridges, leaven uncommunicated for few days several
communities in a radius of 15 km from the volcano.

76         Previous work (Davila et al., 2007; Capra et al., 2010) analyzed the lahars frequency

at Volcán de Colima in relation with the eruptive activity and the characteristics of
rainfalls. Lahars are more frequent at the beginning of the rains season, during short (< 1
hour) stationary rainfalls, with variable rainfall intensities and with only 10 mm of
accumulated rainfall. This behavior has been attributed to a hydrophobic effect of soils on
the volcano slope (Capra et al., 2010). In contrast, in the late rain season, when tropical
rainstorms are common, lahars are triggered depending on the 3-day antecedent rainfall and
with intensities that increase as the total rainfall amount increases (Capra et al., 2010). The
lahars record used for these previous studies was only based on seismic data. Since 2011 a
visual monitoring system have been installed on Montegrande and La Lumbre ravines
(Figure 1), based on which a quantitative characterization of some events (i.e type of flow,
velocity, flow discharge, flow fluctuation) have been possible (i.e. Vázquez et al., 2016a;
Coviello et al., under revision). The aim of the present paper is to better understand the
lahars initiation processes and their dynamical behavior, especially during hurricane events,
when more damages have been observed on inhabited area. In particular, the arrival time of
main lahar's front/surge at the monitoring stations is here analyzed with respect to the
rainfall characteristics (rain accumulation and intensity) in relation with the hydrological



response of the watersheds based on rainfall/infiltration numerical simulation. The
occurrence of discrete surges within lahars have been attributed to spatially and temporally
distributed lahar sources, temporary damming, progressive entrainment of bed material or
change in slope angle (i.e. Iverson 1997; Marchi et al. 2002; Takahashi 2007; Zanuttigh and
Lamberti 2007; Doyle et al., 2010; Kean et al., 2013). Without excluding previous models,
data here presented shows that main pulses within a lahar are not randomly distributed in
time, and they can be correlated with rainfall peak intensity and/or watershed discharge,
depending on the watershed shape and hydrophobic behavior subject to the antecedent soil
moisture. The lahars triggered by the hurricanes Jova, Manuel and Patricia are here used as
they correspond with the best documented events occurred during past years, and they will
be also compared with an extraordinary hydrometeorological event occurred at the begin of
the rain season (11 June, 2013) to better show the drastic change on lahar initiation due to
the hydrophobic effect of soils at Volcán de Colima. Based on rainfall distribution over
time for the analyzed events, a stormwater is here designed, whic can be used to run
simulations prior to an event to have an estimation of the time arrivals of main pulses when
weather forecast is available. The data here presented have important implication for hazard
assessment during extreme hydrometeorological events as a complementary tool of an early
warning system.

2. **Methods and data**
**2.1. La Lumbre and Montegrande watersheds**

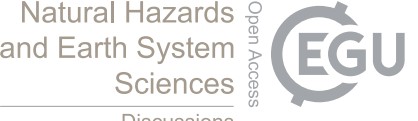



The source area of lahars at Volcán de Colima corresponds to the uppermost unvegetated
portion of the cone (Fig. 1 and 2a), with slopes between 35° and 20°, that also corresponds
with an area of high connectivity, being prone to rills formation and erosive processes
(Ortiz et al., 2017). The channels along main ravines have slopes from 15° up to 4° in the
more distal reach, they are flanked by densely vegetated terraces, up to 15 m high in
average, that consist of debris avalanche and pyroclastic deposits from past eruptions (Figs.
2b and c) (Cortes et al., 2010; Roverato et al., 2011). Seven major watersheds from 2 to 14
km$^2$ feed the main ravines draining from the volcano on the southern side (Fig. 1). La
Lumbre is the largest watershed, with a total area of 14 km$^2$, and Montegrande is in average
with the other catchments, with an area of 2 km$^2$ (Fig. 1). Beside the difference in total
area, the Montegrande and La Lumbre watersheds are quite different in geometry.
Montegrande catchment is elongated, with a maximum width of 800 m, 300 m in average.
In contrast, the proximal portion of the La Lumbre catchment includes all the NW slope of
the cone, to then extent to a more elongated shape towards SW, being up to 1500 m in
width. These differences in area and shape can be correlated with a different response in
water discharge under a rainfall event. In circular drainages, as the proximal portion of the
La Lumbre watershed, all points are quite equidistant from the main river so all the
precipitation reaches the river at the same time, concentrating a large volume of water. In
contrast, in a more elongate basin, lateral drainages quickly drain water on the main
channel at different points but with a lower total discharge. The Gravelius's index Kg
(Bendjoudi and Hubert 2002), which is defined as the relation between the perimeter of the
watershed (P) and that of a circle having a surface equal to that of a watershed (A):

$$Kg = \frac{P}{2\sqrt{\pi A}}$$





is here estimated for Montegrande watershed and for the upper, circular portion of La
Lumbre watershed obtaining values of 1.7 and 1.1 respectively. The lower the value, the
more regular the basin's perimeter and the more prone it is to present high runoff peaks.
Based on these considerations, at La Lumbre watershed a larger volume of water
concentrates along the main channel because of its larger surface and circular shape, but
after a larger period of time respect to Montegrande ravine, where a minor volume of water
quickly reaches the main drainage.

**2.2 Lahar Monitoring at Volcán de Colima**
In 2007, a monitoring program was implemented at Volcán de Colima. At the beginning,
two rain gauges where installed to study lahar initiation (AR and PH sites, Figure 1) and
lahar propagation was detected by the broadband seismic stations of RESCO, the
seismological network of Colima University (Davila et al., 2007; Zobin et al., 2009; Capra
et al., 2010). Afterwards, two monitoring station specifically designed for studying lahar
activity were installed, in 2011 at the Montegrande ravine and in 2014 at La Lumbre ravine
(MSMg and MSL respectively, Figure 1). Both stations consist of a 12 m-high tower with a
directional antenna transmitting data in real time to RESCO facilities, a camcorder
recording images each 2-4 secs with a 704 x 480 pixels in resolution, a rain gauge coupled
with a soil moisture sensor, and a 10 Hz geophone (Vázquez et al., 2016a; Coviello et al.,
under revision). The rain gauge (HOBO RG3) records rain accumulation at one-minute
intervals. At Montegrande ravine seismic data are also obtained from a 3 component Guralp





CMG-6TD broadband seismometer installed at 500 m upstream from the monitoring site,
sampling at 100 Hz (BB-RESCO, Figure 1).
Montegrande station detected lahars occurred during 2011 Jova and 2013 Manuel events,
and lahars triggered during the 2015 Hurrican Patricia were only recorded by La Lumbre
station (Table 1). In fact, in 2011 only MgMS site was operating (as the BB-RESCO
station), and recorded the seismic signal of the lahar associated to Jova and Manuel. No
images are available since both events occurred during the night. The LMS station starts to
operate at the end of 2013 and was able to record the lahars associated to Hurricane Patricia
along the La Lumbre ravine (images and geophone data). In contrast, in 2015 the MgMS
was destroyed by pyroclastic flows during the 10-11 explosive activity, and in October
2015 the new station was still under construction. Only few pictures were acquired and they
are of low quality because of the abundant steam coming from the hot lahars since they
originated from the remobilization of fresh pyroclastic flow deposits (Capra et al., 2016).
The 11 June 2013 event was perfectly captured by the camera installed at the MgMS site
and the BB-RESCO recorded its seismic signal.
The seismic signal is here analyzed to detect the arrival of main flow fronts and discharge
variation. For this, only the amplitude of the signal is considered, which can be correlated
with the variation in the maximum peak flow discharge (Doyle et al., 2010; Vázquez et al.,
2016a). The seismic record is here compared with the available images to identify the main
changes in dynamic of the detected lahars. All the lahars here analyzed correspond to multi-
pulses events as classified by Vazquez et al. (2016a); they consist of long lasting lahars
presenting several pulses each one characterized by a block-rich front followed by the main
body and dilute tail showing continuous changes in flow discharges. A detailed seismic



description of these types of lahars is available in Vázquez et al. (2016a), here we focus on
the number of main flow peaks and their arrival times (Table 2).

**2.3. The hydrometeorological events**
Hurricane Jova formed over the Pacific Ocean, hit the Pacific coast on October 12, 2011, as
a category 2, and traveled inland toward Volcán de Colima. The hurricane arrived as a
tropical storm at the town of Coquimatlán, just 10 km SW of the city of Colima with winds
up to 140 km/h, and 240 mm of rain over 24 h (Fig. 3a). Severe damage was registered in
inhabited area, including the city of Colima where floods damaged roads, bridges and
buildings.
The 2013 Hurricane Manuel of category 1, hit the pacific coast during national holidays
(Fiestas Patria) causing several damage to mountainous region in Guerrero state, triggering
several landslides that caused up to 96 deaths and left several villages uncommunicated as
thousands of tourists trapped at Acapulco and Ixtapa international airports. At Volcan the
Colima rains started on September 15 and lasted for more than 30 hrs with more than 300
mm of accumulated rains (Fig. 3a).
The 2015 Hurricane Patricia was considered as the strongest hurricane on record to affect
Mexico. The system starts to develop on 18 October over the Pacific Ocean, strengthened
into a hurricane shortly after 00:00 GMT 22 October and early on 23 October it reached its
maximum category of 5. But late on the same day, the system rapidly lost its strength. It
landfalls around 23:00 GMT along the coast of the Mexican state of Jalisco near Playa
Cuixmala, about 60 km west-northwest of Manzanillo. On the morning of the 23 October,
2015 it continued to rapidly weaken as it moves on the Sierra Madre Occidental high
relieves. At Colima town, up to 400 mm of rains accumulated along 30 hours since the
morning of 23 October (Fig. 3a). Lahars along the Montegrande ravine were hot since they
originated from the erosion of pyroclastic flow deposits emplaced during the 10-11 July
2015 eruption. Sever damages affected the Colima town and the volcano surrounding. A
bridge along the interstate was destroyed leaving uncommunicated La Becerrara village and
interrupting the traffic between Colima and Jalisco states.
Patricia and Manuel rainfalls show a similar behavior, with a progressive rain accumulation
along 28-30 hrs; in contrast, during Hurricane Jova, 200 mm of rain accumulated in less
than 15 hrs reaching a total of 240 mm during the following 13 hrs (Fig. 3a). These
differences are more evident plotting the 10-min accumulated value normalized over the
total accumulated rainfall (Fig. 3b). Average rainfall intensities calculated over a 10-min
interval range from 32 mm/hrs to 37 mm/hrs for Manuel and Patricia events respectively
and up to 43 mm /hrs for the Hurricane Jova (Table 2). Finally rainfall values were
calculated at selected intervals (15 m, 30 m, 45 mm, 1, 3, 6, 12, 18, 24, 28 hr) to design
possible storm rainfall distributions based on tropical rains associated to hurricanes
recorded so far at Colima Volcano (Table 2). Considering the similar behavior of the
Manuel and Patricia rainfalls, a stormwater can be designed considering their average
values (Fig. 3c) (i.e. NRCS, 2008), based on which a forecast analysis can be performed, as
will be discussed below.

**2.4. Rainfall simulations**



To better understand the lahar behavior and duration during extreme hydrometeorological
event at Volcán de Colima, rainfall simulations were performed with Flo-2D code (O´Brian
et al., 1993). The Flo-2D code routes the overland flow as discretized shallow sheet flow
using the Green-Ampt or the SCS Curve number (or combined) infiltration models. For the
present work the SCS Curve Number (SCS-CN, i.e. Mishra and Singh, 2003) was selected.
With this model, the volume of water runoff produced for the simulated precipitation is
estimated through a single parameter that summarizes the influence of both the superficial
aspects and deep soil, including the saturated hydraulic conductivity, type of land use, and
humidity before the precipitation event. A similar approach was already used for modeling
debris flow initiation mechanisms (i.e. Gentile et al., 2006; Llanes et al., 2015). To apply
the SCS-CN model, it is necessary to classify the soil in one of four groups, each
identifying a different potential runoff generation (A, B, C, D; USDA-NRCS 2007). The
watershed of La Lumbre and Montegrande ravines were subdivided in two main zone: the
unvegetated upper cone and the main channel that consist of unconsolidated pyroclastic
material with large boulders imbedded in sandy to silty matrix, and the vegetated lateral
terraces. Lateral terraces consist of old pyroclastic sequences, with incipient soils and
vegetated with pine trees and sparse brushes, with soils that show a hydrophobic behavior
at the beginning of the rain season (Capra et al., 2010). In-situ infiltration tests were also
performed based on which values of saturated conductivity were obtained in the range of 50
mm/h (nude soil) to 100 mm/h (vegetated) (Ortiz, 2017). Based on these observations, soils
were classified between group A and B (Bartolini and Borselli, 2009). Curve Numbers for
the vegetated terraces and for the nude soils were estimated in 75 and 80 respectively (in
wet season, Hawkins et al., 1985; Ferrer-Julia et al. 2003). To perform simulation with the
FLO-2D code, two polygons were traced to delimit the un-vegetated portion of the cone





from the vegetated area of the watershed, and at each polygon the relative CN value was
assigned. The simulated rain corresponds with the cumulative value calculated at 10
minutes interval (Fig. 3b). At the apex of each watershed a barrier of outflow points were
defined to obtain the total values of the watershed discharge. The simulation was performed
with a 20-m digital elevation model.

**254 3. Results**

During the Jova hurricane, lahars started in Montegrande ravine early in the morning of 12
October, 2011, around 07:20 GMT (here after all time is in GMT), after approximately 40
% of the total rain (240 mm) accumulated (Fig. 4a). The event lasted more than 4 hours,
and three main peaks in amplitude can be detected in the seismic signal (Fig. 4a). In
particular, the first two peaks are similar in amplitude (0.015 cm/s), separated by more than
2 hours of signal fluctuation. After less than one hour from the second peak, a single,
discrete pulse can be recognized (0.05 cm/s in amplitude), followed by a "train" of low-
amplitude seismic peaks that lasted for more than an hour.
Along the same ravine, an extreme event was recorded on 11 June, 2013. This event
corresponds to an extraordinary episode and is here introduced to better discuss the
hydrological response of the Montegrande ravine. It represents an unusual event at the
beginning of the rainy season, considering the total accumulated rainfall of 120 mm in less
than 3 hrs (Table 2), with maximum pick intensity up to 140 mm/hr (Fig. 4b). Based on the
seismic record and the still images of the event, this lahar was previously characterized as a
multi-pulse flow, with three main blocks-rich fronts (I, II and IV, Fig. 4c), with similar



amplitudes (0.015-0.025 cm/s), followed by a main flow body consisting of a homogenous
mixture of water and sediments (with a sediment concentration at the transition between a
debris flow and an hyperconcentrated flow) (III, Fig. 4c) (Vazquez et al. 2016a). The last,
more energetic pulse (0.042 cm/s) was accompanied by a water-rich frontal surge that was
able to reach the lens of the camera (IV, Fig. 4c). Comparing the Jova and the 2013 event
seismic records it is possible to note that in both events, the largest pulse corresponds with
the last one. Flow discharge was estimated for the 2013 event, with a maximum of 120 m$^3$/s
value for the largest pulses (IV, Figure 4b) (Vazquez et al., 2016a). For the Jova event, the
only visual data available are the images of the channel the day before and the day after the
event, where a deep erosion of the channel is visible (Fig. 5), but comparing its seismic
signal with the 2013 lahar, and based on the classification criterion established for lahars at
Volcán de Colima (Vazquez et al., 2016a) each main peak corresponds to the arrival of
flow surges or to block-rich fronts followed by the body of the flow. Fluctuation in seismic
energy along the vertical component reflects variation in flow discharge.
The lahar recorded during the Hurricane Manuel along the Montegrande ravine shows a
similar behavior as described for the Jova event (Fig. 6). As the event occurred during the
night no images are available. Based on the seismic record from the BB-RESCO, lahars
stated around 03:00, and lasted for seven hours. The event was characterized by five main
pulses, which amplitude increases with time (0.012-0.025 cm/s), being the last one the
larger in magnitude (0.04 cm/s). Based on the amplitude values, the first two peaks
correspond to precursory dilute flow waves followed by the three main pulses with block-
rich fronts (I, II and III, Fig 6).



For the Hurricane Patricia seismic data (from the geophone) and still images were recorded
at the La Lumbre monitoring station. Based on these data, at approximately 21:22 a slurry
flow is detected on the main channel (Fig. 7a). First pulses of hyperconcentrated flows were
detected around 01:30 (24 October) which progressively increased in flow discharge and
sediment concentration. Several front waves were observed during flooding (I and II, Fig.
7b) for which an average flow discharge of 80-100 $m^3$/sec was estimated, and two main
pulses arrived at 04:30 and 05:00, with 6 m-depth block-rich fronts and maximum flow
discharge of 900 $m^3$/sec (III, IV, V and VI, Fig. 7b). At around 05:40 the seismic record
detected the arrival of a third pulse. Although no images were available, the amplitude of
the last pulse (0.07 cm/s) suggests it was larger than those previously described. As
observed for the three events recorded at Montegrande ravine, the largest pulse correspond
again with the last one.
The results of rainfall simulations are plotted as a normalized curve of the total discharge,
along with the normalized accumulated rainfall and its intensity (calculated over a 10-min
interval) (Fig. 8). In the same plot, the arrival time of the main lahar pulses here analyzed is
also indicated (red triangles, Fig. 8). By comparing watershed discharge with rainfall
intensity, a general correlation can be observed for the Montegrande basin during Jova and
Manuel hurricane, contrasting with the June 2013 event, where the simulation is not able to
reproduce watershed discharge during the first minutes of the event when most of rainfall is
accumulated and maximum rainfall intensities are detected. For la Lumbre watershed a
clear correlation between peak intensities and watershed discharge is not clearly
observable. If the arrival times of the main lahars' pulses are considered, the events
associated to the hurricanes Jova and Manuel along the Montegrande ravine show a similar



behavior. In both cases early slurry flows are detected after ~40% of the total rain is
accumulated. The main flow pulses better correlate with the highest rain intensity values,
which also correspond with maximum peaks in watershed discharge; the last, largest pulse
corresponds with the maximum peak discharge of the watershed. In contrast, for the
Patricia event, along the La Lumbre ravine, first slurry flows also starts after 40% or
rainfall accumulated, but main lahar pulses fit better with the peaks watershed discharge.
Finally, analyzing the simulation in the Montegrande ravine for the June 2013 event, it is
possible to observe a different behavior. The lahar starts as less than the 10% of rain is
accumulated, and the main lahar pulses perfectly correlate with the peak rainfall intensities,
and only the last largest pulse correlates with the watershed peak discharge.

**4. Discussion**
At present, several attempts to define lahar rainfall thresholds have been already carried out
for different volcanoes (i.e. Lavigne et al., 2000; van Westen and Daag, 2005 Barclay et al.,
2007), including Volcán de Colima (Capra et al., 2010). This study is mostly addressed to
better predict the lahar evolution during extraordinary hydrometeorological event as
hurricane, a common long-duration and large-scale rainfall phenomenon at tropical
latitudes. In particular, we are interested in predicting the arrival of block-rich fronts that
have caused several damages during past events. Based on the seismic and visual data
gathered from the events here analyzed, it is possible to evidence which are the key factors
in controlling the arrival of main lahars fronts. For Jova, Manuel and Patricia events, lahars
started after the 40% of total rain accumulated, and apparently the timing for the initial





pulses correlate well with the peaks of the rainfall intensity for the Montegrande ravine,
while for la Lumbre ravine they better match with the watershed discharge. Nevertheless
for all analyzed cases, the largest pulses correspond with the last ones and correlate with the
peak watershed discharge for all the analyzed examples. The observed difference between
Montegrande and La Lumbre ravines can be correlated with the different areas and shapes
of the two catchments. In fact, due to its elongated shape ($K_G = 1.7$) and small area ($A = 2$
km), the Montegrande watershed shows a quicker response between rainfall and discharge,
with a rapid water runoff that concentrated at different point along the main channel (Fig.
1b). This behavior is much clearer for the June 2013 event, which occurred at the beginning
of the rain season when soils on the lateral terraces of the ravines show a hydrophobic
behavior (Capra et al., 2010). The simulation is not able to reproduce any watershed
discharge at the beginning of the event, because the hydrophobic behavior of the soils
inhibits the infiltration and the water runoff quickly promotes lahar initiation. During this
event, the first lahar pulses perfectly match with the rainfall peak intensities (except for the
last major pulse), starting from the very beginning of the rainfall event. In contrast, La
Lumbre ravine has a wider, rounded upper watershed ($K_G = 1.1$; $A = 14$ km$^2$) that is able to
concentrated a larger volume of water before to turn SW in the main channel where lateral
contribution can still increase water discharge. Even if rainfalls of hurricanes Manuel and
Patricia show a similar behavior (Fig. 3), the catchment response of La Lumbre is clearly
different with a pulsating behavior of lahars mainly controlled by the watershed discharge.
Nevertheless, for all the events here analyzed, the largest pulse corresponds with the last
one recorded and it correlates with the maximum watershed discharge, pointing to a strong
control of the catchments recharge in generating the largest and more destructive pulses.
Previous works correlated the occurrence of surges within a lahar to multiple sources, such



as lateral tributaries along the main channel (i.e. Doyle et al., 2010) or due to the failure of
temporary dams of large clasts in correspondence of an increase in rainfall intensity (Kean
et al., 2013). Lateral tributaries are absent in both Montegrande and La Lumbre channels
and, even if accumulation of clasts it is possible, no significant discontinuities of the
channel bed can be observed upstream the monitoring sites. Based on data here presented,
formation of pulses within a lahar is mostly controlled with the increase in water runoff that
at a critical discharge rate mobilize a large volume of sediment where large clasts
accumulate at its front. This is a well-documented mechanism (i.e. Iverson, 1997), but
based on the model here proposed, the discharge rate is controlled by the watershed
discharge that regulates the timing on the arrival of main pulses, depending on the rainfall
behavior and the watershed shape. Nevertheless, the last pulse always is the largest in
volume. This model is strictly related to migratory, long-duration and large-scale rainfall
events hitting tropical volcanoes such as the Volcán de Colima. In fact, during mesoscale
non-stationary rainfalls, typical at the beginning of the rainy season, lahars are usually
triggered at low accumulated rainfall values and controlled by rainfall intensity due to the
hydrophobic behavior of soils, and they usually consist of uni-pulse events with a single
block-rich front that last less than one hour (i.e. Vázquez et al., 2016b). In perspective, the
results here presented can be used to design an Early Warning System (EWS) for hurricane-
induced lahars, i.e. event triggered by long-duration and large-scale rainfalls. Most
common pre-event or advance-EWSs for debris flows are based on empirical correlations
between rainfall and debris flow occurrence (e.g., Keefer et al., 1987; Aleotti, 2004; Baum
and Godt, 2009). The instruments adopted for debris-flow advance warning are those
normally used for hydrometeorological monitoring and consist of telemetry networks of
rain gauges and/or weather radar. The typical way to represent these relations is identifying



critical rainfall thresholds for debris flow occurrence. The availability of both a large
catalogue of events and a reliable precipitation forecast that could give the predicted
amount of rainfall some hours in advance would allow the issue of an effective warning, at
least in predicting the arrival time of the main lahar pulses. In addition, instrumental
monitoring of in-channel processes can be used to validate a preliminary warning-condition
triggered by wheatear forecast and/or rainfall measurements.

**5. Conclusions**
Real time monitoring of lahars at Volcán de Colima volcanoes reveal that watershed
discharge is the key factor in controlling the arrival of main block-rich fronts during long-
lasting lahar triggered during tropical storms, and that the largest destructive pulses will
arrive after the initial surging. For the 2015 Hurricane Patricia event the weather forecast
predicted an estimated value for the total rainfall, as also the approximate time of its
landfall. Based on the deigned storm obtained with the time rainfall distribution of the
event here analyzed, it could have been possible to anticipate when lahars started along the
La Lumbre ravine, and the arrival time of main pulses. Along the other ravines, that show a
watershed similar to the Montegrande, it could have been possible to predict the arrival of
at least the largest pulse. This information coupled with the real time monitoring could be a
better tool for hazard assessment and risk mitigation. In fact, these findings can be used to
implement an advance warning system based on the monitoring of a hydrometeorological
process to issue a warning before a possible lahar is triggered.





**Acknowledgements.**
This work was supported by CONACyT projects 230 and 220786 granted to Lucia Capra
and by the postdoctoral fellowship of DGAPA (Programa de Becas Posdoctorales de la
UNAM) granted to Velio Coviello. Thanks to José Luis Ortiz and Sergio Rodríguez, from
the Centro de Prevención de Desastres (CENAPRED), who set up the instrumentation on
the Montegrande monitoring site.

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



**Figure captions**

Figure 1. a) Aster image (4, 5 and 7 bands in RGB combination) where main watersheds at Volcán de Colima are represented. The locations of the monitoring stations are indicated. The inset show the location of the raingauge of the Meteorological National Service at the summit of the Nevado de Colima Volcano.

Figure 2. a) Panoramic view of the Volcán de Colima showing the unvegetated main cone mostly composed by loose volcanic fragments. b) Montegrande and c) La Lumbre ravines in the middle reach where it is possible to observe the main channel flanked by 10-15m-high terraces mainly constituted by debris avalanche deposits.

Figure 3. a) Cumulative and b) normalized values of rainfall of hurricanes Jova, Manuel and Patricia calculated at 10 min-intervals. c) Normalized curve of total rainfalls cumulated at 15, 30, 60 minutes and 1, 3, 6, 12, 18, 24, 28 hrs. Dotted line represents the average value between Manuel and Patricia hurricanes.

Figure 4. a) Seismic record of the lahar triggered during the Hurricane Jova, on 12 October, 2011. b) Seismic record of the lahar triggered during the 11 June, 2013 events. Main pulses are indicated with roman letters. c) Images captures by the camera corresponding to the main lahar pulses as indicated in figure b.

Figure 5. Images showing the morphology of the channel at the monitoring site of the Montegrande ravine, a) the day before and b) the day after the Hurricane Jova. c) Topographic profiles showing that the channel was eroded 1.5 m in depth.





Figure 6. Seismic record of the lahar triggered during the Hurricane Manuel, on 15
September, 2013, recorded along the Montegrande ravine
Figure 7. a) Seismic record of the lahar triggered during the Hurricane Patricia, on 26
October, 2015, recorded along the La Lumbre ravine. Main lahar pulses are indicated with
roman letters. b) Images captured by the camera corresponding to the main pulses as
indicated in figure a.
Figure 8. Diagrams showing the main lahar pulses (red triangles) as detected from the
seismic signal of the analyzed events in relation with the accumulated rainfall (dark line),
rainfall intensity (10m/hr) (gray line) and simulated watershed discharge (blue line) for the
following hidrometeorological events a) Jova; b) Manuel; c) 13 June, 2013; and d) Patricia.
Table 1. Data collected for the events here studied.

Table 2. Normalized accumulated rains (in percentage) at progressive time steps.



Table 1. Data collected for the events here studied.

| Event | ravine | Seismic record | Image record | Total rain (mm) | Max. rain intensity (mm/hr) |
|---|---|---|---|---|---|
| Jova | Montegrande | | | 240 | 43 |
| Manuel | Montegrande | X | | 300 | 32 |
| Patricia | Lumbre | X | X | 400 | 37 |
| 11 June 2013 | Montegrande | X | X | 120 | 140 |







Table 2. Normalized accumulated rains (in percentage) at progressive time steps.

| Event/time (hrs) | 0.25 | 0.5 | 1 | 2 | 3 | 6 | 12 | 24 | 27 |
|---|---|---|---|---|---|---|---|---|---|
| **Jova** | 0.0011 | 0.0016 | 0.0035 | 0.0172 | 0.0329 | 0.1411 | 0.7073 | 0.968 | 0.9943 |
| **Manuel** | 0.0023 | 0.0035 | 0.0042 | 0.0072 | 0.0151 | 0.0341 | 0.1548 | 0.735 | 0.9181 |
| **Patricia** | 0.0002 | 0.0004 | 0.0009 | 0.0062 | 0.0174 | 0.0556 | 0.2544 | 0.829 | 0.9782 |
| *average* | *0.00125* | *0.00195* | *0.00255* | *0.0067* | *0.01625* | *0.04485* | *0.2046* | *0.782* | *0.9481* |

The average values refer to hurricanes Manuel and Patricia.

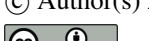



Fig. 01

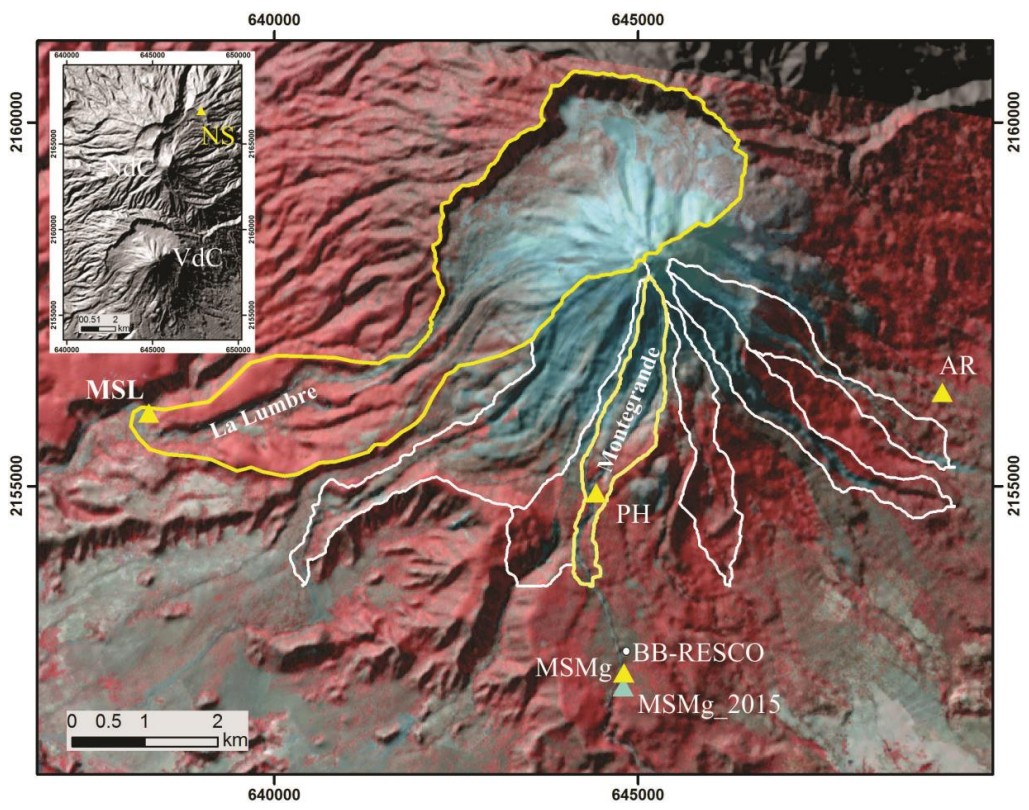





Fig. 02

 





Fig- 03





Fig. 04

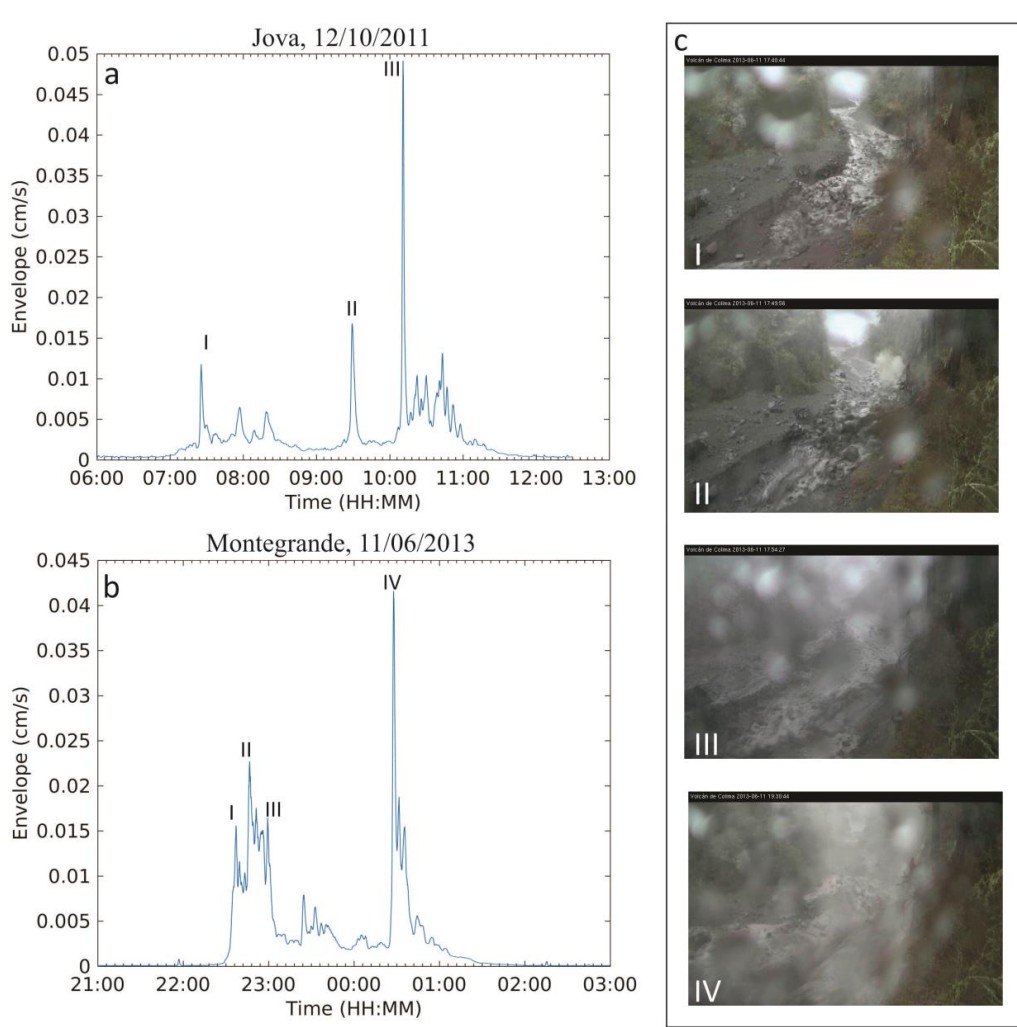





Fig. 05

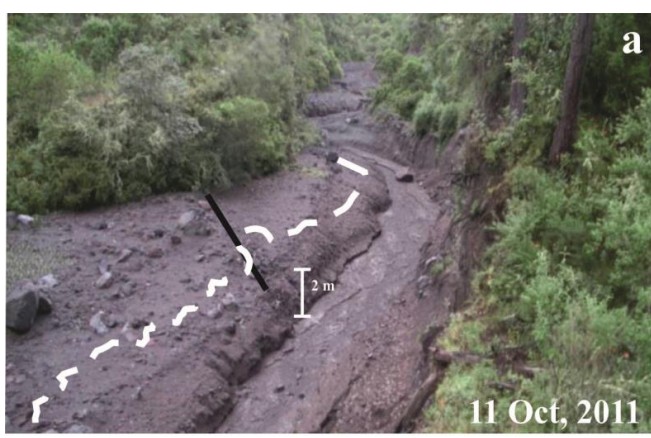

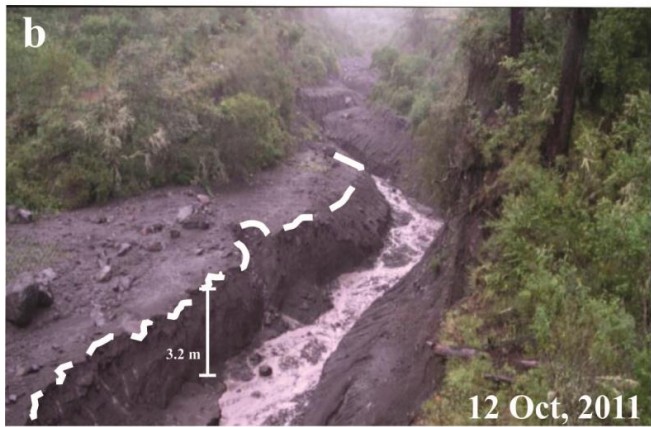

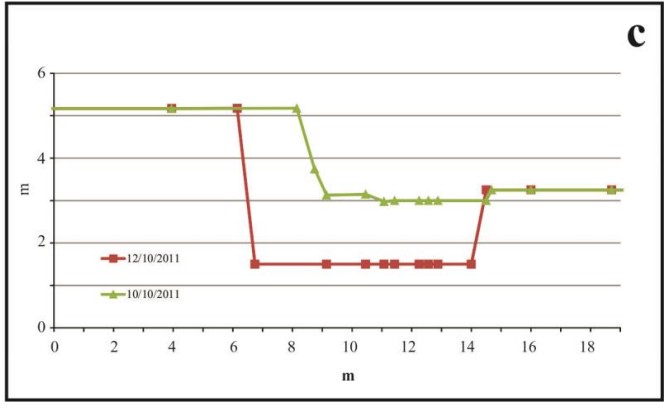



Fig. 06

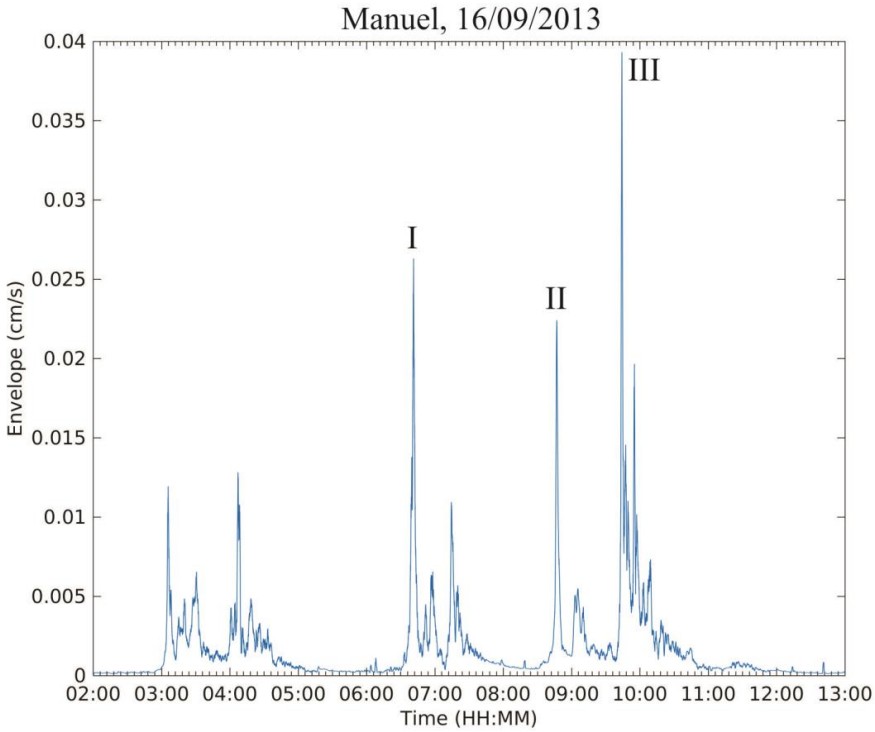






Fig. 07

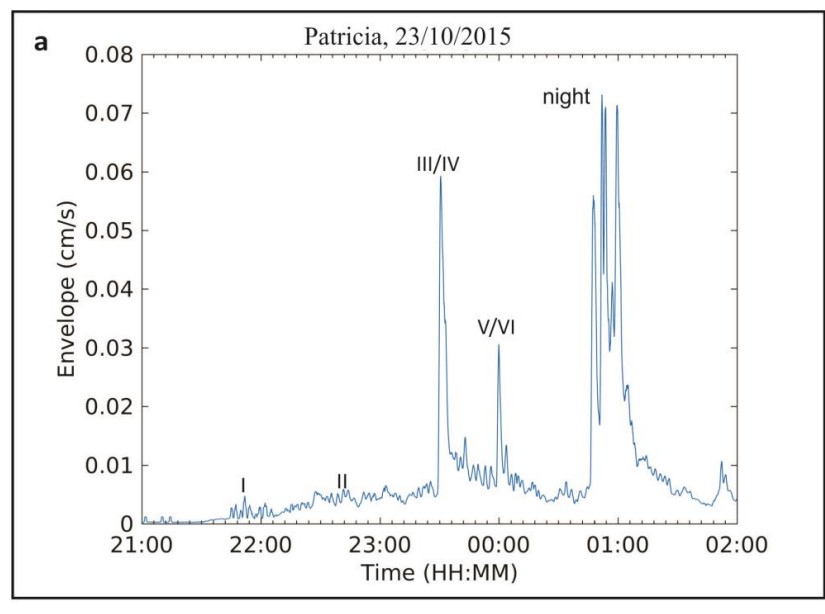

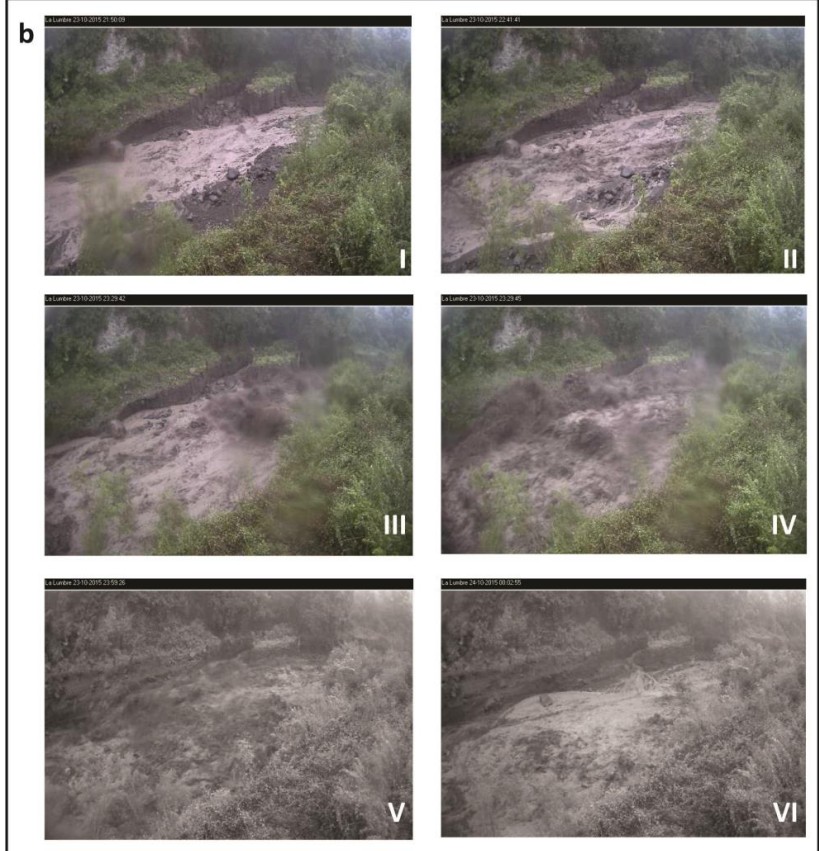





Fig. 08

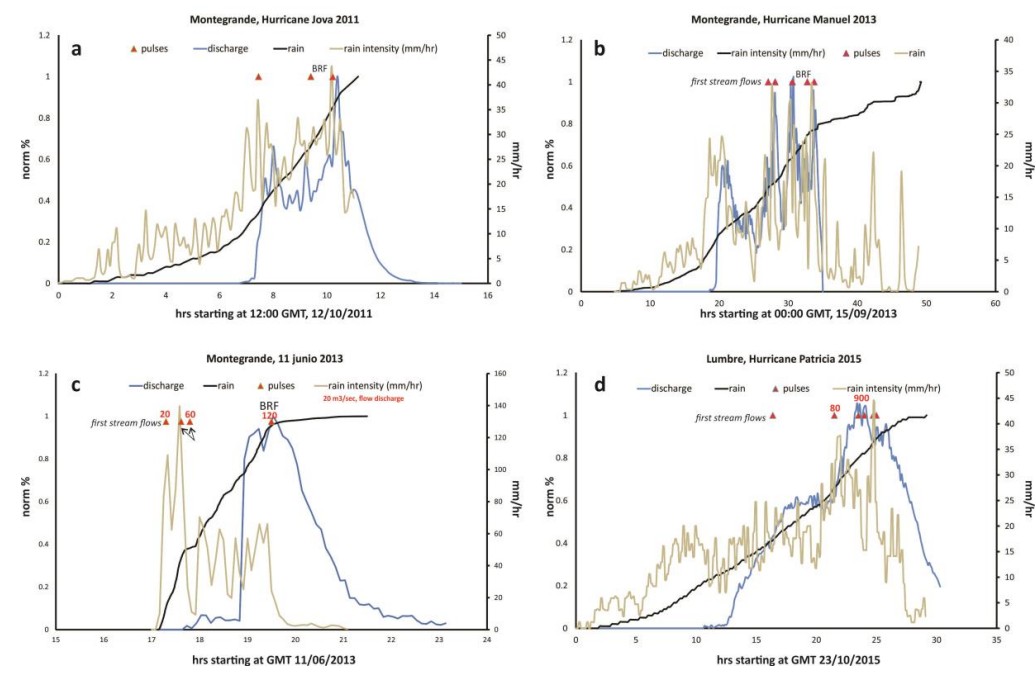

