# Peer review of "runoff modelling, seismic a"

_Natural Hazards and Earth System Sciences, 2017_

## Short Comment (SC1) · 27 Oct 2017

The paper of Lucia Capra and her colleagues provides a valuable contribution to the analysis of the relationships between flood runoff formation and lahar occurrence during hurricanes. Lahar monitoring and characterization of hydraulic properties of soils in a difficult environment deserve to be stressed. The aim of this note is to propose some comments on specific aspects of the analysis.

The core of the study is the assessment of the runoff response to hurricanes and the comparison of simulated flood hydrographs with the monitored lahars. Since no measurements of water discharge are available in the studied catchments, rainfall-runoff

modeling (this term should be preferred to "rainfall simulation") remains essentially uncalibrated. It is well-known that a careful selection of model parameters does not ensure a satisfactory correspondence between simulated and real hydrographs. The lack of rainfall-runoff model calibration and the impossibility of performing it in the studied catchments should be acknowledged and discussed. More could be said, moreover, about the propagation of rainfall excess computed by means of the SCS Curve Number method: this part of runoff simulation is of utmost importance for the timing of flood response. A sensitivity analysis on rainfall-runoff model parameters, although does not surrogate model calibration, could help coping with the uncertainties in the assessment of flood response.

The impossibility of calibrating rainfall-runoff models is the reason why simulated water flood hydrographs have seldom been compared with observed debris flow hydrographs in catchments instrumented for debris flow monitoring.

A possible, even if only partial, check of model results with the observed runoff response could consist in the comparison of the time of the first rise of the simulated hydrograph with video images showing the onset of the water flood at the monitoring stations. According to figure 8, this comparison could be possible for Hurricane Manuel at Montegrande (Fig. 8b) and Hurricane Patricia at La Lumbre (Fig. 8d), whereas the early occurrence of lahars prevents it in the other two cases (Figs. 8a and 8c).

---

## Referee Comment (RC1) · Anonymous Referee #1 · 31 Oct 2017

article Review of the paper.

30 Oct 17

The paper provides an interesting study about the relationship between the rain induced by hurricanes and the generation of lahars.

The paper mostly requires an English grammar revision. Nevertheless, I suggest that as the Coulomb failure criterion was not mentioned in the paper, to include it within the paper, perhaps when the authors mention landslide triggering empiric criterion (section Discussion).

It draws attention that in the abstract, numerical modeling of rain and infiltration is promised. None of them are fulfilled. The O'Brian model is a shallow water approach for surface flows, despite the claim done by the authors within the paper that it was used for rain fall modeling.

In addition, there are few more suggestions listed bellow.

**1 Abstract**

Review English

**2   Methods and data**

1. line 132: use primary source (Gravelius, 1914)

2. line 175. Review English.

3. Line 224: Mistake, the aim of Flo2D is not to do rainfall simulations.

4. Line 228: clarify how do you simulated the precipitation.

5. Line 235: zones

**3   Results**

1. Line 278, figure 5: keep the previously used convention for the sub-figure numbering (top left hand side).

2. Line 305: English

3. Line 321. English

**4   Discussion**

1. Line 333: English

2. Line 352: English

3. Line 356: English

4. Line 354: English

5. Line 392: English

6. Line 397: English

7. Line 398: English

8. Line 400: if actually "it could have been possible" , why it was not possible? It is always risky to extrapolate, thus to advise extrapolations.

---

## Short Comment (SC2) · 9 Nov 2017

Dr Lucia Capra Centro de Geociencias Universidad Nacional Autónoma de México (UNAM) Campus Juriquilla Queretaro MÉXICO

9th November 2017

Re: NHESS 2017-354

Dear Dr Capra

I have now had the opportunity to complete my review of your manuscript "Hydrological control of large hurricane-induced lahars: evidence from rainfall, seismic and video

monitoring' submitted to the journal NHESS. Overall I find this to be an interesting paper suitable for inclusion in the journal. I have made numerous annotations on an attached hardcopy of the manuscript, mainly related to matters of English style and grammar, in addition to the following numbered points. More clarification is needed on the nature of the rainfall-runoff simulation model used.

0. There is no mention of the catchment rainfall-runoff model simulations in the title of the paper, yet these are a significant part of the manuscript. 1. How do you define lahar size? By peak discharge, and if so where? Or by peak seismic amplitude by using this as a proxy for lahar volumetric discharge, even though the seismic energy output of a lahar is a function of many factors including volumetric discharge, sediment concentration and sediment grain-size distribution. 2. This sentence is unclear, there appear to be some key words missing. Some kind of couple catchment rainfall-runoff simulation model is being invoked. 3. Hurricanes and cyclones are not globally distributed. 3A. Mt Ruapehu is not a tropical volcano, despite its rich rain-triggered lahar record. 4. Insert the full date. 6. Insert the exact date. The Fiestas Patria will have no meaning outside of Mexico. 6A. Move the underlined text up to the *. 7. This sentence reads like there are three zones, unless you are combining the channel and terraces into one. Clarify please. 7A. Move this sentence to line 173. 8. Move the underlined text down to line 316. 9. Move the indicated block of text to line 316 before the insertion (8) above. 10. A critical weakness of using the 40% of total rainfall threshold is that it is difficult to know when this point has been reached when it is still raining, unless you have a great deal of faith in your weather forecasts. Do you have accurate predicted total rainfall and distribution curves for these events that could be run through your simulator and compared with the actual lahar events? 11. This implies that there is no lag time between the peak rainfall intensity measured 6 km away on another volcanic edifice and the arrival of the lahar peak at the detectors. 12. How long does it take to run Flo-2d, could it be run in real-time by feeding in the incoming rainfall intensity data? 13. Clarify. 14. So the simulation cannot duplicate the initial hydrophobic behaviour? 15. I'm assuming that these catchments are ungauged, so there is no way of calibrat-

ing the simulated discharge produced by the rainfall-runoff routing model? 16. I'm not sure what this third graph c) is showing. 17. Give a little more detail about how this envelope (cm/s) plot is derived. 18. Ignore my scribbles on this figure.

Yours sincerely Dr Vern Manville University of Leeds

Please also note the supplement to this comment:
https://www.nat-hazards-earth-syst-sci-discuss.net/nhess-2017-354/nhess-2017-354-SC2-supplement.pdf

**Supplement:**

*@ No mention of catchment rainfall-runoff model in title.?*

[revised manuscript text omitted]

better predict *~ions of* the lahar evolution during extraordinary hydrometeorological event*s* *such* as hurricane*s*, a common long-duration and large-scale rainfall phenomenon ^*in* at tropical *~i not extra ordinary* ⑬

latitudes. In particular, we are interested in predicting the arrival of block-rich ^*flow* fronts that have caused *seven*  damage*s* during past events. Based on the seismic and visual data gathered from the events here analyzed, it is possible to evidence *identify* which  the key factors in controlling the *timing* arrival of main lahar*s* *the* fronts. For Jova, Manuel and Patricia events, lahars started after the 40% of total rain accumulated, and apparently the timing for the initial *had* [※] ^

pulses correlate*s* ^well with the peaks of the rainfall intensity for the Montegrande ravine, *T* ⑪

while for la Lumbre ravine they better match with the *peak simulated* ⑫ ^watershed discharge. Nevertheless for all analyzed cases, the largest pulses correspond with the last ones and correlate with the peak *simulated* ^watershed discharge for all the analyzed examples. The observed difference between

⑩

[※] *I have to predict when the 40% point is reached when it is still raining .....*

⑪ *T no lag time??*

⑫ *Is it possible to run Flo-2D in real-time*

[revised manuscript text omitted]

Figure 01

[Figure]

Montegrande ravine

La Lumbre ravine

Figure 02

[Figure]

Figure 3

Figure 4

Jova, 12/10/2011

Montegrande, 11/06/2013

[Figure]

[Figure]

[Figure]

Figure 5

[Figure]

[Figure]

Figure 7

[Figure]

Figure 08

---

## Referee Comment (RC2) · Anonymous Referee #2 · 15 Nov 2017

**Summary**

The role of rainfall characteristics (e.g. distribution, duration, volume and intensity) and catchment properties (e.g. size, shape) in determining lahar initiation and the downstream arrival of lahar pulses is an area of important study, particularly at tropical volcanoes. In this manuscript *Capra et al.* study rain-triggered lahars at Colima Volcano in two catchments to (in total) four different rainfall events. The two catchments are the *Montegrade* watershed, a long catchment with limited width and, the *La Lumbre* watershed, a large, semi-spherical catchment north-north-west of Colima. Using rainfall, geophone, and image records collected during Hurricanes Jova, Manuel, Patricia

and an extreme rain event (in June 2013). Geophone records, correlated with imagery, for the Montegrade catchment identified the arrival of lahar pulses. In all cases (Jova, Manuel and 11 June), the largest peak occurred last and coincided with the arrival of a block-rich lahar front. Similar behaviour was observed in the La Lumbre watershed during Hurricane Patricia.

These observations were augmented by 'rainfall simulations' of the event in Montegrade and La Lumbre catchments. My main critique of this work surrounds the use of these simulations, model assumptions and conclusions drawn from this model that may not be fully justified. Despite this, the authors present some interesting analysis and conclusions to the problem of rainfall-induced lahar occurrence which, subject to the following modifications, is well suited to this journal.

**Main issues**

As mentioned above, the rainfall simulations used in this work need to be clarified and care needs to be taken when analysing and drawing conclusions from the simulation results. In particular:

1. **What are the assumptions of the SCS curve model and how may it affect results?**

   The SCS approach is a simplified method for estimating rainfall runoff. Lower curve numbers result in less runoff for the same amount of rainfall. However, as stated on lines 229-231, this model simplifies the complex relationship between rainfall and overland flow into a single number. A weakness of this approach is that the curve number does not consider the effects of single storm properties (e.g. rainfall intensity) on infiltration.

2. **How was rainfall applied over the simulation domain?**

The authors state that the rainfall 10 minute intervals were applied to the simulation (lines 249-50). However, there is no indication if this varied spatially. If a spatially homogeneous rainfall input was used, the authors need to indicate this and, in discussion, consider the effect of this assumption on results and implication for the migratory, long duration rainfall scenarios.

3. Related to point 1, in Fig. 8, simulated discharge shows better correlation to identified lahar pulses during Hurricanes Jova, Manuel and Patricia. In these events, rainfall intensity is much lower and cumulative rainfall is more linear than the 11 June event. This highlights a potential limitation of the runoff erosion model that needs to be identified and discussed.

4. Although correlation between observed lahar pulses and simulated discharge indicate a level of agreement between simulation and reality, the models have not been calibrated to real world (i.e. measured discharge) data. In effect, the model can then only *indicate* differences in watershed response between the Montegrade and La Lumbre catchments.

Based on these issues, elements of the discussion and conclusion may need modification:

Line 338: pulses better match *simulated* watershed discharge. This is a crucial distinction, as without calibration we cannot estimate the potential error in the discharge rate.

Line 338-340: "Nevertheless ...", in Fig. 8c, only one of the four observed pulses coincide with the simulated discharge - this correlation could be (in my opinion likely is) pure coincidence for this event - you need to account for this. I would recommend removing this sentence entirely, as it is largely repeated in lines 357-359.

Line 368-371: "This is a well documented mechanism ..." it is hard to interpret what is being said here. What is the difference between discharge rate and watershed discharge? How does one control the other? Rainfall intensity and watershed shape seem to control the arrival of main pulses more than discharge.

Overall, I suggest to the authors that the strength of this manuscript is in the correlations of multiple streams of data (rainfall intensity, cumulative rain, geophone records) to examine the relationship between rainfall and lahar pulses. Since the rainfall simulations are uncalibrated, they add some context to the discussion, but simulation results (in their current form) cannot be used to draw conclusions about the relationship. I believe the manuscript would be greatly improved by a rewording of the discussion, reducing the emphasis on rainfall simulations and instead focusing on the relationship between rainfall characteristics and lahar pulses.

**Technical and minor issues**

Please see the attached .pdf for corrections to English style and grammar.

Line 38, 160, 219: What is a 'stormwater'? This is unclear terminology

Line 58: Ruapehu is not in a tropical region.

Line 161, 165, 170/Figure 1: "MgMS" do you mean MSMg?

Line 163/Figure 1: "LMS" do you mean MSL?

Line 193/194: Change to "Volcán de Colima"

Line 202/203: "Sierra Madre Occidental high relieves" perhaps just Sierra Madre Occidntal range?

Line 225: Reference is O'Brien et al.

Line 317-318 and 320: See above discussion, I think it is important to state the pulses match with peak *simulated* discharge.

Line 322-324: Given model assumptions and disparities when compared to the other events, there is a high chance this correlation is coincidental. If you want to note the correlation here, you should also highlight the disparity.

Line 333-335: Reword sentence to fix grammar... Seismic and visual data from events analysed here provide evidence to key factors...

Line 338-380 and 357-359: See above, these two sentences are almost exactly the same. Recommend removing the first instance.

Line 398-399: "Based on the deigned storm obtained..." meaning is unclear, be specific on the requirements to anticipate start time and arrival of lahar pulses.

Fig. 1 caption: "...locations of the monitoring stations are indicated by triangles"

Fig. 1: Is station MSMg_2015 identified in the manuscript? If not, remove.

Fig. 3b/c: As a normalised plot, there is no need for the 'y' (norm) axis to be greater than one. Adjust to be between 0 and 1.

Fig. 5c is unnecessary, remove.

Fig. 8 needs to be improved, suggest the following:

- In the caption, rain intensity is a gray line, but in the figure it is gold/yellow.

- Fig. 8b - "Rain" and "Rain intensity" legend entries are switched

- Left axis (%norm) should only be between 0 and 1 (see above)

- Arrows in Fig. 8c do not seem to indicate anything - should "first stream flows" text be placed nearby?

- Color and line choice makes it hard to discriminate between rain intensity and discharge. Try adjust colors or line thicknesses.

Table 1: The manuscript suggested 'Jova' had seismic records for Montegrade ravine?

Please also note the supplement to this comment:
https://www.nat-hazards-earth-syst-sci-discuss.net/nhess-2017-354/nhess-2017-354-RC2-supplement.pdf

**Supplement:**

[revised manuscript text omitted]

Fig. 01

[Figure]

[Figure]

Fig. 02

Montegrande ravine

La Lumbre ravine

[Figure]

[Figure]

Fig- 03

Fig. 04

[Figure]

[Figure]

[Figure]

Fig. 05

[Figure]

[Figure]

[Figure]

[Figure]

[Figure]

Fig. 06

[Figure]

[Figure]

Fig. 07

[Figure]

[Figure]

[Figure]

[Figure]

Fig. 08

[Figure]

---

## Editor Comment (EC1) · S. Segoni (Editor) · 18 Nov 2017

Dear authors, even if the open discussion is still open and others short comments may be posted, all referees' reports have been received. Therefore, if you are interested to speed up the revision process, I invite you to start working on the answers to the comments and on the revised version of the manuscript. I recommend you to take into consideration ALL comments received, as the reports highlight several points that need to be better addressed before publication.

Regards,

[Figure]

S.

---

## Author Comment (AC1) · 17 Jan 2018

Responses to L. Marchi. We would like to thank the reviewer for the comments and constructive suggestions made to improve the present work. Please find below the reviewer's comment and authors' replies to these comments.

The paper of Lucia Capra and her colleagues provides a valuable contribution to the analysis of the relationships between flood runoff formation and lahar occurrence during hurricanes. Lahar monitoring and characterization of hydraulic properties of soils in a difficult environment deserve to be stressed. The aim of this note is to propose some comments on specific aspects of the analysis. The core of the study is the assessment of the runoff response to hurricanes and the comparison of simulated flood hydrographs with the monitored lahars. Since no measurements of water discharge are available in the studied catchments, rainfall-runoff modeling (this term should be preferred to "rainfall simulation") remains essentially uncalibrated. It is well-known that a careful selection of model parameters does not ensure a satisfactory correspondence between simulated and real hydrographs. The lack of rainfall-runoff model calibration and the impossibility of performing it in the studied catchments should be acknowledged and discussed. More could be said, moreover, about the propagation of rainfall excess computed by means of the SCS Curve Number method: this part of runoff simulation is of utmost importance for the timing of flood response. A sensitivity analysis on rainfall-runoff model parameters, although does not surrogate model calibration, could help coping with the uncertainties in the assessment of flood response. The impossibility of calibrating rainfall-runoff models is the reason why simulated water flood hydrographs have seldom been compared with observed debris flow hydrographs in catchments instrumented for debris flow monitoring. A possible, even if only partial, check of model results with the observed runoff response could consist in the comparison of the time of the first rise of the simulated hydrograph with video images showing the onset of the water flood at the monitoring stations. According to figure 8, this comparison could be possible for Hurricane Manuel at Montegrande (Fig. 8b) and Hurricane Patricia at La Lumbre (Fig. 8d), whereas the early occurrence of lahars prevents it in the other two cases (Figs. 8a and 8c).

-We perfectly agree with the reviewer. As pointed out, no measurements of water discharge are available at both La Lumbre and Montegrande watershed, so a model calibration is not possible. We followed the suggestion by L. Marchi and we calibrated the simulated watershed discharge using the information gathered from video images acquired by the monitoring station of La Lumbre ravine during the Patricia event. For Montegrande ravine a calibration would be possible only for the 11 June 2013 event, but considering the strong effect of soil hydrophobicity at the beginning of the rainy season it is difficult to set up a comparison. For the new version of the manuscript, a rainfallrunoff modeling was performed with both SCS-CN and Green-Ampt (G-A) methods. We decided to also run the simulations with the G-A infiltration method to discuss the limitation of the SCS-CN that does not consider the rainfall intensity (for more detail see response to RC2). The simulated watershed discharge obtained with the G-A method best fits with the initial shallow-water flow observed in the video images, however, main peaks discharges corresponding with the main lahars pulses are equally reproduced with both infiltration models (see new figure R1 at the end of this document). Based on this result, and considering the limited number of parameters needed to apply the SCS-CN method, we focused on the latter method that would be more suitable to adopt in an early warning system devoted to forecast the lag time of main lahar pulses at a specific site. We improved and modified the section "2.4. Rainfall-runoff modelling" as follows (see also response to RC2). Other authors already performed a sensitive analysis of the G-A method, showing that the saturated hydraulic conductivity Ks is a key factor in the estimation of infiltration rates and exerts a notable influence on runoff calculations (i.e. Chen et al., 2015). With respect to the SCS-CN model, the only input parameter is the Curve Number, thus we present a simple comparison for Patricia event at La Lumbre ravine. Results obtained with the 80/75 CN values for channel and vegetated area respectively are compared with two other simulations performed using global values of 75 and 80 (see table R2). This exercise shows that the uncertainty in simulated maximum peak discharge is in the range of 0.1 hr, pointing that a global CN value could be also used for the Volcán de Colima. Here below the new section for the paper.

-2.4. Rainfall-runoff modelling

To better understand the lahar behavior and duration during extreme hydrometeorological events at Volcán de Colima, rainfall-runoff simulations were performed with Flo-2D code (O'Brian et al., 1993). The Flo-2D code routes the overland flow as discretized shallow sheet flow using the Green-Ampt or the SCS Curve number (or combined) infiltration models. For the present work, the SCS Curve Number (SCS-CN, i.e. Mishra

and Singh, 2003) was selected but a comparison between both infiltration models is presented below. The rainfall is applied to the entire watershed, without spatial variability as we are dealing with large-scale, long-duration hurricane-induced rainfall. This rainfall is discretized as a cumulative percent of the total precipitation each 10 minutes. With the SCS-CN model, the volume of water runoff produced by the simulated precipitation is estimated through the use of a single parameter, i.e. the Curve Number (CN). This parameter summarizes the influence of both the superficial and deep soil features, including the saturated hydraulic conductivity, type of land use, and humidity before the precipitation event (for an accurate description of the origin of the method see Rallison, 1980; Ponce and Hawkins, 1996). A similar approach was previously used for modeling debris flow initiation mechanisms (i.e. Gentile et al., 2006; Llanes et al., 2015). To apply the SCS-CN model, it is necessary to classify the soil in one of four groups, each identifying a different potential runoff generation (A, B, C, D; USDA-NRCS 2007). La Lumbre and Montegrande watersheds were subdivided into two main zones: 1) the unvegetated upper cone and the main channel, that consists of unconsolidated pyroclastic material with large boulders embedded in a sandy to silty matrix, and 2) the vegetated lateral terraces, composed by old pyroclastic sequences with incipient soils and are vegetated with pine trees and sparse bushes. Based on these observations, soils were classified between group A and B (Bartolini and Borselli, 2009). CN for the vegetated terraces and for the nude soils is estimated at 75 and 80 respectively (in wet season, Hawkins et al., 1985; Ferrer-Julia et al. 2003). To perform a simulation with the FLO-2D code, two polygons were traced to delimit the un-vegetated portion of the cone from the vegetated area of the watershed, and at each polygon the relative CN value was assigned. At the apex of each watershed a barrier of outflow points were defined to obtain the values of the simulated watershed discharge computed at each 0.1 hr. The simulation was performed with a 20-m digital elevation model. One of the limitations of the SCS-CN model is that it does not consider the effect of the rainfall intensity on the infiltration. In addition, since no measurements of water discharge are available at both La Lumbre and Montegrande basins, it is difficult to calibrate the simulations

here presented. To investigate the SCS-CN model uncertainties in the assessment of flood response, the Green-Ampt (1911) model (G-A), sensitive to the rainfall intensity, was also applied and results were compared with the outcome of SCS-CN model. For the G-A method, the main input parameters are the saturated hydraulic conductivity (Ks), the soil suction and the volumetric moisture deficiency. Ks is the key factor in the estimation of infiltration rates and exerts a notable influence on runoff calculations, therefore it requires great care in its measurement (Grimaldi et al., 2013). These values can be extrapolated from reference tables or directly measured with field experiments. Based on the textural characteristics of soils at Volcán de Colima as well as type of vegetation, input parameters were selected from the FLO-2D reference manual. In particular, with a value of Ks of 20 mm/hr the simulated watershed discharge best fits with the precursory shallow-water flow observed in the video images, as it will be showed below (Figure R1). The Ks value of 20 mm/hr is equivalent to the CN value used for the SCS-NC simulation. In fact an empirical relation between Ks and CN has been proposed be Chong and Teng (1986): $S = 3.579(Ks1)^{1.208}$ where S is the potential retention and it is related to the CN as follow (Mockus, 1972): $CN = 2540/(S + 25.4)$ Based on these equations, a value of Ks equal to 20 mm/hr corresponds to a CN of 75.5 in the range of values here used for the SCS-NC infiltration model.

The G-A infiltration model was tested at La Lumbre ravine, using the Patricia rainfall and comparing the simulated watershed discharge curve with the available video images. Figure R1 shows the discharge curve that best fits with the data gathered from the images (Table #), based on which the two method were qualitative calibrated. The G-A infiltration model nicely reproduce the initial scouring of a muddy water and it corresponds with the first increase in the simulated watershed discharge. The SCS-CN infiltration model is not able to reproduce this first water runoff. This can be explained considering that the initial abstraction due to the interception, infiltration and surface storage, is automatically computed in the SCS-NC model as 0.2S, being probably too high for the studied area. In contrast, with the G-A method, the initial abstraction can be modified and best results were obtained with a value of 6 mm that corresponds to

a surface typical of a vegetated mountain region (Table #). However, both infiltration models give similar results for the main peaks of the simulated maximum watershed discharge that correspond with the arrival of the main lahar pulses as observed from the image (Figure R1). These results show that the G-A model is much more reliable to detect precursory slurry flows, while both models are equally able to catch the main surges of a lahar. One important point is that the simulations are here used to set up an early warning system to forecast the lag time of the main lahar surges. The first slurry flows were here important to calibrate the G-A simulation but they do not represent an essential data for the early warning system. In addition, input data for the G-A method often are difficult to set, requiring great care in its measurement; in contrast, the output of the SCS-CN method only depends on the CN value. The SCS-CN method has been largely used in rainfall-runoff modeling, and we consider that is a valuable method for the objective of the present work, as we are not seeking for a quantitative estimation of the watershed discharge but on the arrival time of the main lahar pulses.

Additional references

Li Chen, Long Xiang, Michael H. Young, Jun Yin, Zhongbo Yu, Martinus Th. van Genuchten, 2015. Optimal parameters for the Green-Ampt infiltration model under rainfall conditions. J. Hydrol. Hydromech., 63, 2015, 2, 93–101 Chong, S. K., and Teng, T. M. (1986). "Relationship between the runoff curve number and hydrologic soil properties." J. Hydrol., 84(1–2), 1–7. Mishra, S. K., and Singh, V. P. (2003). Soil conservation service curve number (SCS-CN) methodology, Kluwer Academic Publishers, Dordrecht, Netherlands. Mockus, V. (1972). Estimation of direct runoff from storm rainfall national engineering handbook, Soil Conservation Service, Washington, DC. Ponce, V., and Hawkins, R. (1996). "Runoff curve number: Has it reached maturity?" J. Hydrol. Eng., 10.1061/(ASCE)1084-0699(1996)1:1(11), 11–19.

Figure R1. Comparison of simulated watershed discharge curves based on SCS-NC and G-A infiltration models. Qualitative calibration is here proposed based on the flow discharge as observed at the MSL site.

**Fig. 1.** Figure R1

*Table R1*

| Parameter used in the G-A simulations | |
|---|---|
| *Abstraction* | 6 (mm) |
| *Ks* | 20 (mm/hr) |
| *soil-suction* | 100 (mm) |
| *initial saturation* | 0.35 |
| *final saturation* | 0.7 |

*Table R2. SCS-CN simulations with different CNs*

| Surges observed in the images | peak III (23.5 hr) | peak IV (24 hr) |
|---|---|---|
| **CN** | *time in the simulated watershed discharge curve* | |
| 75 global | 23.4 | **24.1** |
| 80/75 (channel/vegetated) | **23.5** | **24.1** |
| 80 global | **23.5** | 24.2 |

**Fig. 2.** Table R1 and R2

---

## Author Comment (AC2) · 17 Jan 2018

Response to RC1. We would like to thank the reviewer for the comments and constructive suggestions made to improve the present work. Please find below the reviewer's comment and authors' replies to these comments.

The paper provides an interesting study about the relationship between the rain induced by hurricanes and the generation of lahars. The paper mostly requires an English grammar revision. Nevertheless, I suggest that as the Coulomb failure criterion was not mentioned in the paper, to include it within the paper, perhaps when the authors mention landslide triggering empiric criterion (section Discussion).

[Figure]

-We consider that the Coulomb failure criterion is out of the focus of the present paper, we are not discussing the condition of lahar initiation; lahars at Volcan de Colima originate from a progressive erosion of material from the river bed.

It draws attention that in the abstract, numerical modeling of rain and infiltration is promised. None of them are fulfilled. The O'Brian model is a shallow water approach for surface flows, despite the claim done by the authors within the paper that it was used for rain fall modeling.

-We agree with the reviewer and we were wrongly using the terminology, in fact the paper presents rainfall-runoff simulations, as also point out by the SC1.

In addition, there are few more suggestions listed below.

-We took into account of the following suggestions. The English revision was based on the suggestions made by RC2 and SC2.

1 Abstract

Review English

2 Methods and data

1. line 132: use primary source (Gravelius, 1914)

done

2. line 175. Review English.

3. Line 224: Mistake, the aim of Flo2D is not to do rainfall simulations.

Changed to rainfall-runoff simulation

4. Line 228: clarify how do you simulated the precipitation.

This is now clarified as follow.

The rainfall is applied to the entire watershed, without a spatial variation, and it is

discretized as a cumulative percent of the total precipitation each 10 minutes.

5. Line 235: zones

done

3 Results

1. Line 278, figure 5: keep the previously used convention for the sub-figure numbering (top left hand side).

done

8. Line 400: if actually "it could have been possible" , why it was not possible? It is always risky to extrapolate, thus to advise extrapolations.

This refers that if at the time of Patricia event this model was ready, the simulation could have been run to have a forecast of the arrival times of the main lahar surges. The text was slightly modified as follow

For the 2015 Hurricane Patricia event the weather forecast predicted an estimated value for the total rainfall, and also the approximate time of its landfall. Based on the deigned storm obtained with the rainfall/time distribution of the analyzed events, it would have been possible to anticipate when lahars started along the La Lumbre ravine, and the arrival time of main pulses. Then, this first prediction could be constrained using rainfall-runoff modeling based on real-time monitoring data, as simulations do not take more than 30 minutes to run.

---

## Author Comment (AC3) · 17 Jan 2018

Responses to V. Manville.

We would like to thank the reviewer for the comments and constructive suggestions made to improve the present work. Please find below the reviewer's comment and authors' replies to these comments.

Title. The reviewer suggests to mention rainfall-runoff simulation into the title. -We agree and we modified it as follow: "Hydrological control of large hurricane-induced lahars: evidences from rainfall-runoff modellin, seismic and video monitoring"

[Figure]

1(line 31). How do you define lahar size? By peak discharge, and if so where? Or by peak seismic amplitude by using this as a proxy for lahar volumetric discharge, even though the seismic energy output of a lahar is a function of many factors including volumetric discharge, sediment concentration and sediment grain-size distribution.

R1: Yes, we used the amplitude as a proxy for lahar volumetric discharge. On previous published works at Volcán de Colima (Vazquez et al., 2016), the size of lahars has been classified based on their seismic response (amplitude, validated with image data) and duration. With available images, the maximum pick discharge was calculated and assigned to the maximum amplitude recorded form the seismic station. We agree that it is not always possible to correlate the amplitude of the seismic signal with the flow depth, but based on real time data gathered at Colima, there is a quite good correlation for those large events (See. Fig. 5 Vazquez et al., 2016). The figure below extracted from Vazquez et al., 2016, clearly point to a correlation between lahar amplitude and flow discharge.

To better state this concept we slightly change the text at line # 183 as follow:

In particular, for lahars at Volcán de Colima a correlation between the maximum peaks in amplitude and the maximum peaks in flow discharge was found (Fig. 5 in Vázquez et al., 2016). Fluctuation in seismic energy along the vertical component reflects variation in flow discharge.

2 (line 37). This sentence is unclear, there appear to be some key words missing. Some kind of couple

R2: Here we refer that based on rainfall data of Manuel and Patricia hurricanes, which show a very similar behavior, a "synthetic" rainfall curve has been designed (in accumulated percentage). If the amount of rain can be estimated prior to an event, this curve could be used to run a rainfall-runoff simulation to try to have a possible forecast. The sentence was modified as it:

A theoretical rainfall distribution curve was here designed based on the rainfall/time distribution of hurricanes Manuel and Patricia. Then, weather forecasts can be used to run simulations prior to the actual event, in order to estimate the arrival times of main pulses, usually characterized by block-rich fronts, which are responsible for most of damage to infrastructures and loss of goods and lives.

3(line 44). Hurricanes and cyclones are not globally distributed.

R3: We modified the sentence as suggested:

"In recent years hurricanes have had catastrophic effects on volcanoes in the tropics troughs the triggering of lahars (sediment-water gravity-driven flows on volcanoes)."

3A(line 55). Mt Ruapehu is not a tropical volcano, despite its rich rain-triggered lahar

R4: The Mt. Ruapehu reference was deleted.

4 and 6 (line 164 and 188). Insert the full date.

R4 and 6: The full date for Patricia and Manuel date of landfalls were added. The sentence was modifies as follow:

In contrast, in 2015 the MgMS site was destroyed by pyroclastic flows during the 10-11 July explosive activity, and in October 2015 the new station was still under construction. Hurricane Manuel (category 1), hit the Pacific coast on 15 September 2013 causing several damage to mountainous region in Guerrero state, triggering several landslides that caused up to 96 deaths and left several villages cut of, as while thousands of tourists were trapped at Acapulco and Ixtapa international airports.

6A (line 200). The sentence was modified as suggested.

R6A: Hurricane Patricia on 2015 was considered as the strongest hurricane on record to affect Mexico. The system began to develop on 18 October over the Pacific Ocean, strengthened into a hurricane shortly after 00:00 GMT on 22 October and early on 23 October it reached its maximum category of 5, before losing strength as it moved

onto the Sierra Madre Occidental range. Landfalls occurred around 23:00 GMT on 23 October along the coast of the Mexican state of Jalisco near Playa Cuixmala, about 60 km west-northwest of Manzanillo.

7(line 234). This sentence reads like there are three zones, unless you are combining the channel and terraces into one. Clarify please.

R7: The sentence was clarified: The watershed of La Lumbre and Montegrande ravines were subdivided into two main zones: 1) the unvegetated upper cone and the main channel that both consist of unconsolidated pyroclastic material with large boulders embedded in a sandy to silty matrix, and 2) the vegetated lateral terraces.

7A (line 279). Move this sentence to line 173.

R7A: This sentence was moved as suggested (se answer to point 1)

8 and 9 (line 311-329). Move the underlined text down to line 316 and move the indicated block of text to line 316 before the insertion.

R8 and 9: The sentece was modified as follow:

Finally, analyzing the simulation in the Montegrande ravine for the 11 June 2013 event, it is possible to observe a different behavior. The lahar starts as less than the 10% of the total rain is accumulated, and the main lahar pulses perfectly correlate with the peak rainfall intensities, and only the last largest pulse correlates with the watershed peak discharge. For la Lumbre watershed, in 2015 a clear correlation between peak rainfall intensities and simulated watershed discharge is not clear. For the Patricia event, along the La Lumbre ravine, first slurry flows also starts after 40% or total rainfall, but main lahar pulses fit better with the simulated peaks watershed discharge.

10. A critical weakness of using the 40% of total rainfall threshold is that it is difficult to know when this point has been reached when it is still raining, unless you have a great deal of faith in your weather forecasts. Do you have accurate predicted total rainfall and distribution curves for these events that could be run through your simulator and

compared with the actual lahar events?

R10: We agree with the reviewer. Here we are only pointing to the evidences get from data here presented (not from the simulations!) that after 40% of the total rainfall first lahars are detected for all the analyzed events. This corresponds to an amount of accumulated rainfall of 100, 120 and 160 mm of rain for Jova, Manuel and Patricia respectively. This evidence points that after at least 100 mm of rains had accumulated (measured in real time from raingauges) lahars can occur. The early warning system will be based on rainfall-runoff modeling results. For the Patricia event the trajectory and time of landfall was quite well predicted, and data about the amount of rainfall were also provided. The text was modified as follow:

For the Jova, Manuel and Patricia events, lahars started after the 40% of total rain had accumulated (corresponding to c. 100, 120 and 160 mm of rain respectively), and apparently the timing for the initial pulses correlates well with the peaks of the rainfall intensity for the Montegrande ravine, while for La Lumbre ravine they better match with the peak simulated watershed discharge.

11 (line 335). This implies that there is no lag time between the peak rainfall intensity measured 6 km away on another volcanic edifice and the arrival of the lahar peak at the detectors.

R11: As observed for the Hurricane Jova, rainfall data from the station at Montegrande and La Lumbre ravine are almost identical (more than 8 km away). This means that the rainfall behavior is quite constant over a large area during a hurricane. Similar behavior is observed for Patricia event, by comparing the Nevado station with the raingauge at Ciudad de Colima. So even if data here used for the Hurricane Patricia are from a station located 6 km away from Volcan de Colima, we are considering that the rainfall intensity was quite homogeneous over these two volcanoes. The Figure R2 (see below) will be added as an extra panel to Fig. 3.

12. How long does it take to run Flo-2d, could it be run in real-time by feeding in the

incoming rainfall intensity data?

R 12: For the simulation here performed, using a 20 m DEM in resolution, each simulation took no more than 30 minutes at our facility so yes, it could be possible to run simulation in real time as data are acquired.

13. Clarify. R13: The phrase was slightly modified as follow:

The observed difference between Montegrande and La Lumbre ravines can be correlated with the different areas and shapes of the two catchments. In fact, due to its elongated shape (KG = 1.7) and small area (2 km2), the Montegrande watershed shows a quicker response between rainfall and discharge, with a rapid water concentration at different point along the main channel (Fig. 1b).

14. So the simulation cannot duplicate the initial hydrophobic behaviour?

R14: No, with the parameter here used, even changing the SCS to 95% (almost impermeable) the simulation was not able to reproduce water discharge at the time the lahars were detected. This is probably again related with the initial abstraction that is fixed by the program based on the CN value (see comment below and responses to reviewer RC2).

15. I'm assuming that these catchments are ungauged, so there is no way of calibrating the simulated discharge produced by the rainfall-runoff routing model?

R 15: Yes the reviewer is correct, direct measurement of watershed discharge is not available. Also based on the comments by the other reviewers we added a section to try to validate the simulation using the video images recorded by La Lumbre monitoring station. Apparently first stream flows are detected at the same time the simulated watershed discharge curve increases. Please refer to response to reviewer RC2 for more detail on this point.
* * *
2017-354, 2017.

**Fig. 5** Plots of the data recorded during the 11 June 2013 lahar from 17:30 to 19:00 (local time). **a** Envelope of the AFM-T signal (the seismic amplitude is in counts). **b** Comparative plot where the *green line* describes the discharge curve in m³/s and the related values of superficial velocity in m/s (*yellow line*), both obtained from the recorded images. *Black arrows* indicate the main features of the flow, as observed in Fig. 4

**Fig. 1.** Figure 5 from Vazquez et al 2016

[Figure]

Fig. R2. Normalized rainfall of the Jova and Patricia event as gathered form different stations, pointing to a quasi-stationary rainfall behavior.

**Fig. 2.** Figure R2

---

## Author Comment (AC4) · 17 Jan 2018

Response to RC2.

We would like to thank the reviewer for the comments and constructive suggestions made to improve the present work. Please find below the reviewer's comment and authors' replies to these comments.

Main issues As mentioned above, the rainfall simulations used in this work need to be clarified and care needs to be taken when analysing and drawing conclusions from the simulation results. In particular:

[Figure]

1. What are the assumptions of the SCS curve model and how may it affect results? The SCS approach is a simplified method for estimating rainfall runoff. Lower curve numbers result in less runoff for the same amount of rainfall. However, as stated on lines 229-231, this model simplifies the complex relationship between rainfall and overland flow into a single number. A weakness of this approach is that the curve number does not consider the effects of single storm properties (e.g. rainfall intensity) on infiltration.

R1: We agree with the reviewer that SCS-NC method does not consider the effect of the rainfall intensity on infiltration, a key point for the cases here analyzed. But it is worth mentioning that here the rainfall input for the FLO-2D simulation is given as a no-linear hydrograph curve where accumulated rainfall is discretized at each 10 minutes interval (as detected with the raingauge). Based also on the comment by L. Marchi (SC1), we tested the Green-Ampt (G-A) rainfall-infiltration method and we calibrated it with the images available for the Patricia event along the La Lumbre ravine, at least for the arrival time of the first slurry flow and for the main surges (this last correlation was already presented in fig.8). The parameters used for the Green-Ampt method were selected from FLO-2D reference tables according to the textural characteristics of the soil on the watershed (Table R1). The $K_s$ (saturated hydraulic conductivity) of 20 mm/hr gives the best fit, and based on the equation proposed by Chong and Teng (1986) it corresponds to a CN of 75.5 in the range of the value used for the simulation performed with the SCS-NC method (see detailed explanation in the text below). It is worth to mention that the input parameters here used for the G-A model represent an average value for the entire watershed.

Figure R1 (see below) shows the comparison of the discharge curve obtained with the SCS and G-A methods and their comparison with selected images of the flow along the La Lumbre channel during the Patricia event. One first issue is the coincidence of the first water runoff along the channel observed in the image with the rise of the discharge in the curve modeled with G-A method, as the SCS-CN is not able to reproduce it. In fact, we performed additional simulation to try to reproduce the initial slurry flow with the SCS method but it was impossible. This can be explained considering that the model automatically assumes an initial abstraction (rainfall intercepted by vegetation) of 0.2S, where S is the potential retention included in the CN calculation (CN=2540/(S+25.4)) (Mockus, 1972), value that it is too high for the studied area. In contrast, the value of initial abstraction can be controlled performing the simulations with the G-A method. However, the main peak discharges corresponding with the main lahar pulses are equally reproduced with both models. Under this evidence, we are able to affirm that the G-A method is much more reliable to detect the first streamflow, but the SCS method is also able to catch the main surges. One important point is that the simulations are here used to set up an early warning system to forecast the lag time of main lahar pulses at a specific site. The first water runoff along the channel was fundamental to calibrate the G-A simulation but it is not an essential data for the early warning system. In addition, input data for the G-A method are probably much more difficult to set, in contrast to the SCS method where only one parameter is needed. A new section has been added within the paragraph "2.4. Rainfall-runoff modeling" to show the comparison between the two infiltration methods based on which the SCS model was selected to be used in the early warning system. The SCS method has been largely used in rainfall-runoff estimations, and we consider that is a valuable method for the objective of the present work. This section was modified as follow:

2.4. Rainfall-runoff modelling To better understand the lahar behavior and duration during extreme hydrometeorological events at Volcán de Colima, rainfall-runoff simulations were performed with Flo-2D code (O'Brian et al., 1993). The Flo-2D code routes the overland flow as discretized shallow sheet flow using the Green-Ampt or the SCS Curve number (or combined) infiltration models. For the present work, the SCS Curve Number (SCS-CN, i.e. Mishra and Singh, 2003) was selected but a comparison between both infiltration models is presented below. The rainfall is applied to the entire watershed, without spatial variability as we are dealing with large-scale, long-duration hurricane-induced rainfall. This rainfall is discretized as a cumulative percent of the total precipitation each 10 minutes. With the SCS-CN model, the volume of water runoff produced by the simulated precipitation is estimated through the use of a single parameter, i.e. the Curve Number (CN). This parameter summarizes the influence of both the superficial and deep soil features, including the saturated hydraulic conductivity, type of land use, and humidity before the precipitation event (for an accurate description of the origin of the method see Rallison, 1980; Ponce and Hawkins, 1996). A similar approach was previously used for modeling debris flow initiation mechanisms (i.e. Gentile et al., 2006; Llanes et al., 2015). To apply the SCS-CN model, it is necessary to classify the soil in one of four groups, each identifying a different potential runoff generation (A, B, C, D; USDA-NRCS 2007). La Lumbre and Montegrande watersheds were subdivided into two main zones: 1) the unvegetated upper cone and the main channel, that consists of unconsolidated pyroclastic material with large boulders embedded in a sandy to silty matrix, and 2) the vegetated lateral terraces, composed by old pyroclastic sequences with incipient soils and are vegetated with pine trees and sparse bushes. Based on these observations, soils were classified between group A and B (Bartolini and Borselli, 2009). CN for the vegetated terraces and for the nude soils is estimated at 75 and 80 respectively (in wet season, Hawkins et al., 1985; Ferrer-Julia et al. 2003). To perform a simulation with the FLO-2D code, two polygons were traced to delimit the un-vegetated portion of the cone from the vegetated area of the watershed, and at each polygon the relative CN value was assigned. At the apex of each watershed a barrier of outflow points were defined to obtain the values of the simulated watershed discharge computed at each 0.1 hr. The simulation was performed with a 20-m digital elevation model. One of the limitations of the SCS-CN model is that it does not consider the effect of the rainfall intensity on the infiltration. In addition, since no measurements of water discharge are available at both La Lumbre and Montegrande basins, it is difficult to calibrate the simulations here presented. To investigate the SCS-CN model uncertainties in the assessment of flood response, the Green-Ampt (1911) model (G-A), sensitive to the rainfall intensity, was also applied and results were compared with the outcome of SCS-CN model. For the G-A method, the main input parameters are

the saturated hydraulic conductivity (Ks), the soil suction and the volumetric moisture deficiency. Ks is the key factor in the estimation of infiltration rates and exerts a notable influence on runoff calculations, therefore it requires great care in its measurement (Grimaldi et al., 2013). These values can be extrapolated from reference tables or directly measured with field experiments. Based on the textural characteristics of soils at Volcán de Colima as well as type of vegetation, input parameters were selected from the FLO-2D reference manual. In particular, with a value of Ks of 20 mm/hr the simulated watershed discharge best fits with the precursory shallow-water flow observed in the video images, as it will be showed below (Figure R1). The Ks value of 20 mm/hr is equivalent to the CN value used for the SCS-NC simulation. In fact an empirical relation between Ks and CN has been proposed be Chong and Teng (1986): $S=3.579Ks^{1.208}$ where S is the potential retention and it is related to the CN as follow (Mockus, 1972): $CN=2540/(S+25.4)$ Based on these equations, a value of Ks equal to 20 mm/hr corresponds to a CN of 75.5 in the range of values here used for the SCS-NC infiltration model. The G-A infiltration model was tested at La Lumbre ravine, using the Patricia rainfall and comparing the simulated watershed discharge curve with the available video images. Figure R1 shows the discharge curve that best fits with the data gathered from the images, based on which the two method were qualitative calibrated. The G-A infiltration model nicely reproduce the initial scouring of a muddy water and it corresponds with the first increase in the simulated watershed discharge. The SCS-CN infiltration model is not able to reproduce this first water runoff. This can be explained considering that the initial abstraction due to the interception, infiltration and surface storage, is automatically computed in the SCS-NC model as 0.2S, being probably too high for the studied area. In contrast, with the G-A method, the initial abstraction can be modified and best results were obtained with a value of 6 mm that corresponds to a surface typical of a vegetated mountain region. However, both infiltration models give similar results for the main peaks of the simulated maximum watershed discharge that correspond with the arrival of the main lahar pulses as observed from the image (Figure R1). These results show that the G-A model is much more reliable to detect

precursory slurry flows, while both models are equally able to catch the main surges of a lahar. One important point is that the simulations are here used to set up an early warning system to forecast the lag time of the main lahar surges. The first slurry flows were here important to calibrate the G-A simulation but they do not represent an essential data for the early warning system. In addition, input data for the G-A method often are difficult to set, requiring great care in its measurement; in contrast, the output of the SCS-CN method only depends on the CN value. The SCS-CN method has been largely used in rainfall-runoff modeling, and we consider that is a valuable method for the objective of the present work, as we are not seeking for a quantitative estimation of the watershed discharge but on the arrival time of the main lahar pulses.

2. How was rainfall applied over the simulation domain? The authors state that the rainfall 10 minute intervals were applied to the simulation (lines 249-50). However, there is no indication if this varied spatially. If a spatially homogeneous rainfall input was used, the authors need to indicate this and, in discussion, consider the effect of this assumption on results and implication for the migratory, long duration rainfall scenarios.

R2: The rainfall was applied to the entire watershed, no spatial variation was assumed. As stated before, the total amount of accumulated rainfall is discretized in 10 minutes interval, introduced in the code as a no-linear hydrograph. During tropical rainfalls rains are nearly stationary on top of the volcano. This can be observed by comparing rainfall data from different stations (fig R2). This figure will be added as extra panel in figure 3.

3. Related to point 1, in Fig. 8, simulated discharge shows better correlation to identified lahar pulses during Hurricanes Jova, Manuel and Patricia. In these events, rainfall intensity is much lower and cumulative rainfall is more linear than the 11 June event. This highlights a potential limitation of the runoff erosion model that needs to be identified and discussed.

R3: The 11 June 2013 event is presented to stress the fact that at the beginning of

the rains season no-stationary, orographic events trigger lahars after few minutes of accumulated rainfall (∼10 mm); in those cases, main pulses are clearly controlled by rainfall peak intensities, mainly because of a strong hydrophobic effect of the soils (see Capra et al., 2010). Therefore, the model here presented does not work for such type of events and can be only used during tropical rains associated to hurricanes, with low rainfall intensities and long durations. This concept is clearly stated in the discussion:

This model is strictly related to long-duration and large-scale rainfall events hitting tropical volcanoes such as the Volcán de Colima. In contrast, during mesoscale non-stationary rainfalls, typical at the beginning of the rainy season, lahars are usually triggered at low accumulated rainfall values and manly controlled by rainfall intensity due to the hydrophobic behavior of soils, and they usually consist of single-pulse events with one block-rich front that last less than one hour (i.e. Vázquez et al., 2016b). In perspective, the results presented here can be used to design an Early Warning System (EWS) for hurricane-induced lahars, i.e. event triggered by long-duration and large-scale rainfalls.

4. Although correlation between observed lahar pulses and simulated discharge indicate a level of agreement between simulation and reality, the models have not been calibrated to real world (i.e. measured discharge) data. In effect, the model can then only indicate differences in watershed response between the Montegrade and La Lumbre catchments. Based on these issues, elements of the discussion and conclusion may need modification:

R4. We totally agree. However, based on the calibration presented in the new section we consider that the model here used is reliable. Yes, Montegrande and La Lumbre have a different watershed response, which clearly controls the arrival time of the main lahar pulses that can be simulated with the rainfall-runoff modeling here proposed.

Line 338: pulses better match simulated watershed discharge. This is a crucial distinction, as without calibration we cannot estimate the potential error in the discharge

rate.

-Again, we think that this aspect is now better justified with the new information based on the comparison between G-A and SCS methods.

Line 338-340: "Nevertheless ...", in Fig. 8c, only one of the four observed pulses coincide with the simulated discharge - this correlation could be (in my opinion likely is) pure coincidence for this event - you need to account for this. I would recommend removing this sentence entirely, as it is largely repeated in lines 357-359.

-As stated into the test the 11 June 2013 event does not fit with the model here proposed, but apparently only the last largest pulse correspond with the simulated watershed discharge.

Line 368-371: "This is a well documented mechanism ..." it is hard to interpret what is being said here. What is the difference between discharge rate and watershed discharge? How does one control the other? Rainfall intensity and watershed shape seem to control the arrival of main pulses more than discharge.

-We agree we the reviewer and we simplified this section as follow.

Based on data presented here, formation of pulses within a lahar is mostly controlled by the watershed shape that regulates the timing of the arrival of main pulses, depending on the rainfall behavior. Nevertheless, the last pulse is always the largest in volume.

Overall, I suggest to the authors that the strength of this manuscript is in the correlations of multiple streams of data (rainfall intensity, cumulative rain, geophone records) to examine the relationship between rainfall and lahar pulses. Since the rainfall simulations are uncalibrated, they add some context to the discussion, but simulation results (in their current form) cannot be used to draw conclusions about the relationship. I believe the manuscript would be greatly improved by a rewording of the discussion, reducing the emphasis on rainfall simulations and instead focusing on the relationship between rainfall characteristics and lahar pulses.

-Base on the reviewers' comments and the comparison between the SC-NC and G-A infiltration models, we consider that at present our model is much more well justified. Simulations represent an important issue for the present work and, as proposed here, they can be used to perform an early warning system at least to determine the time arrivals of main lahars pulses.

Technical and minor issues Please see the attached .pdf for corrections to English style and grammar. All the suggestion to English style and grammar were taken into account.

Line 38, 160, 219: What is a 'stormwater'? This is unclear terminology This expression was changed to "theoretical rainfall distribution curve"

Line 58: Ruapehu is not in a tropical region. It was also observed by SC", so this example was removed

Line 161, 165, 170/Figure 1: "MgMS" do you mean MSMg? Yes, it is now corrected.

Line 163/Figure 1: "LMS" do you mean MSL? Yes, now corrected

Line 193/194: Change to "Volcán de Colima" Done

Line 202/203: "Sierra Madre Occidental high relieves" perhaps just Sierra Madre Occidntal range? Also based to the SC2 reviewer, the sentence was changes. The system began to develop on 18 October over the Pacific Ocean, strengthened into a hurricane shortly after 00:00 GMT on 22 October and early on 23 October it reached its maximum category of 5, before losing strength as it moved onto the Sierra Madre Occidental range.

Line 225: Reference is O'Brien et al. Done

Line 317-318 and 320: See above discussion, I think it is important to state the pulses match with peak simulated discharge. Also based on SR2, the text was clarified.

Line 322-324: Given model assumptions and disparities when compared to the other

events, there is a high chance this correlation is coincidental. If you want to note the correlation here, you should also highlight the disparity. We consider that as already stated into the text, the 11 june 2013 event is here reported only to show the different watershed response at the beginning of the rain season. The model here proposed will be not used to predict the arrival of main pulses for the events at the beginning of the rain season.

Line 333-335: Reword sentence to fix grammar... Seismic and visual data from events analysed here provide evidence to key factors... Also based on SC2 comment, the sentence was changes as follow: Based on the seismic and visual data gathered from the events analyzed here, it is possible to identify the key factors in controlling the arrival timing of main lahar fronts.

Line 338-380 and 357-359: See above, these two sentences are almost exactly the same. Recommend removing the first instance. We agree and 338-380 lines were deleted.

Line 398-399: "Based on the deigned storm obtained..." meaning is unclear, be specific on the requirements to anticipate start time and arrival of lahar pulses.

For the 2015 Hurricane Patricia event the weather forecast predicted an estimated value for the total rainfall, and also the approximate time of its landfall. Based on the deigned storm obtained with the rainfall/time distribution of the analyzed events, it would have been possible to anticipate when lahars started along the La Lumbre ravine, and the arrival time of main pulses. Then, this first prediction could be constrained using rainfall-runoff modeling based on real-time monitoring data, as simulations do not take more than 30 minutes to run.

Fig. 1 caption: "...locations of the monitoring stations are indicated by triangles" Done

Fig. 1: Is station MSMg_2015 identified in the manuscript? If not, remove. The station is now included into the text.

Fig. 3b/c: As a normalised plot, there is no need for the 'y' (norm) axis to be greater than one. Adjust to be between 0 and 1. Done

Fig. 5c is unnecessary, remove. Done

Fig. 8 needs to be improved, suggest the following: • In the caption, rain intensity is a gray line, but in the figure it is gold/yellow. • Fig. 8b - "Rain" and "Rain intensity" legend entries are switched • Left axis (%norm) should only be between 0 and 1 (see above) • Arrows in Fig. 8c do not seem to indicate anything - should "first stream flows" text be placed nearby? • Color and line choice makes it hard to discriminate between rain intensity and discharge. Try adjust colors or line thicknesses.

Figure was improved as suggested (see next page)

Table 1: The manuscript suggested 'Jova' had seismic records for Montegrade ravine? Yes, corrected.

Please also note the supplement to this comment: https://www.nat-hazards-earth-syst-sci-discuss.net/nhess-2017-354/nhess-2017-354- RC2-supplement.pdf

All suggested changes were done

Additional References

Li Chen, Long Xiang, Michael H. Young, Jun Yin, Zhongbo Yu, Martinus Th. van Genuchten, 2015. Optimal parameters for the Green-Ampt infiltration model under rainfall conditions. J. Hydrol. Hydromech., 63, 2015, 2, 93–101

Chong, S. K., and Teng, T. M. (1986). "Relationship between the runoff curve number and hydrologic soil properties." J. Hydrol., 84(1–2), 1–7.

Mishra, S. K., and Singh, V. P. (2003). Soil conservation service curve number (SCS-CN) methodology, Kluwer Academic Publishers, Dordrecht, Netherlands.

Mockus, V. (1972). Estimation of direct runoff from storm rainfall national engineering

handbook, Soil Conservation Service, Washington, DC.

Ponce, V., and Hawkins, R. (1996). "Runoff curve number: Has it reached maturity?" J. Hydrol. Eng., 10.1061/(ASCE)1084-0699(1996)1:1(11), 11–19.

[Figure]

Figure with axes labeled % (vertical) and hrs (GMT) (horizontal), comparing Green-Ampt and SCS curves, main surges, with photographs below.

Images no available
for the last two pulses

Initial water runoff

First surge detected in the
seismic record

First main pulse

Second main pulse

Figure R1. Comparison of simulated watershed discharge curves based on SCS-NC and G-A infiltration models. Qualitative calibration is here proposed based on the flow discharge as observed at the MSL site.

**Fig. 1.** Figure R1

*Table R1*

| Parameter used in the G-A simulations | |
|---|---|
| *Abstraction* | 6 (mm) |
| *Ks* | 20 (mm/hr) |
| *soil-suction* | 100 (mm) |
| *initial saturation* | 0.35 |
| *final saturation* | 0.7 |

*Table R2. SCS-CN simulations with different CNs*

| Surges observed in the images | peak III (23.5 hr) | peak IV (24 hr) |
|---|---|---|
| **CN** | *time in the simulated watershed discharge curve* | |
| **75 global** | 23.4 | 24.1 |
| **80/75 (channel/vegetated)** | 23.5 | 24.1 |
| **80 global** | 23.5 | 24.2 |

**Fig. 2.** Table R1 and R2

[Figure]

Fig. R2. Normalized rainfall of the Jova and Patricia event as gathered form different stations, pointing to a quasi-stationary rainfall behavior.

**Fig. 3.** Figure R2

**Montegrande, Hurricane Jova 2011**

▲ pulses    — discharge    — accum. rain    — rain intensity (mm/hr)

a

norm % / mm/hr

BRF

hrs starting at 12:00 GMT, 12/10/2011

**Montegrande, Hurricane Manuel 2013**

▲ pulses    — discharge    — cum. rain    — rain intensity (mm/hr)

b

*first slurry flows*    BRF

norm % / mm/hr

hrs starting at 00:00 GMT, 15/09/2013

**Montegrande, 11 junio 2013**

— discharge (mm/hr)    — accum. rain    ▲ pulses    — rain intensity (mm/hr)

c

*first stream flows*    20   60    120 BRF    20 m3/sec, flow discharge

norm % / mm/hr

hrs starting at GMT 11/06/2013

**Lumbre, Hurricane Patricia 2015**

— discharge    — accum. rain    ▲ pulses    — rain intensity (mm/hr)

d

80 ▲    900

norm % / mm/hr

hrs starting at GMT 23/10/2015

Figure 08

**Fig. 4.** Figure 8 modified

[Figure]